# North Pacific Subtropical Mode Water Volume and Density Anomalous Decrease in a Recent Stable Kuroshio Extension Period from Argo Observations

Jing Sheng[1,2], Cong Liu[1], Yanzhen Gu[1,2], Peiliang Li[1,2], Fangguo Zhai[3], Ning Zhou[1,2]

[1]Hainan Institute of Zhejiang University, Sanya, 572000, China.
[2]Ocean College, Zhejiang University, Zhoushan, 316000, China
[3]College of Oceanic and Atmospheric Science, Ocean University of China, Qingdao, 266000, China.

*Correspondence to*: Cong Liu (liucong175@gmail.com) Yanzhen Gu (guyanzhen@zju.edu.cn)

**Abstract.** North Pacific subtropical mode water (NPSTMW) is formed as the low stratification water mass in the wintertime mixed layer south of the Kuroshio Extension (KE). In a recent period of 2018−2021, the KE jet is in a persisting stable dynamic state. But based on analysis of Argo observation, the mean volume of NPSTMW in ventilation region drop anomalously by ~21% during 2018−2021 relative to 2012−2015 when the KE jet is likewise stable. Moreover, the NPSTMW volume in denser density range (approximately $\sigma_\theta$ >25.2 kg m$^{-3}$) starts to decrease since 2018. The decreasing of the NPSTMW subduction and formation rate are associated with anomalously shallow wintertime mixed layer depth (MLD) and weak heat loss in the NPSTMW formation region. The diagnosis in ML heat budget indicates that the wintertime upper ocean warming is the key factor in determining the recent MLD decrease in the NPSTMW formation region. The decrease of air-sea heat exchange and the vertical entrainment act to weaken the vertical mixing and decrease the MLD, resulting in the weakening of subduction. The interannual variations in the air-sea heat exchange and vertical entrainment reflect the variability of the overlying atmosphere which is correlated with Pacific decadal oscillation (PDO) shift in 2018−2021. When the PDO shifts from its positive to a negative phase in analysis period, the effects of local wind stress anomalies seem to play an evident role in driving the variability of NPSTMW on interannual time scales. The MLD and heat loss change during the cold season in 2018−2021 are strongly coupled with the poleward shift of the westerlies−which cause the weaker wintertime wind and the easterly wind anomalies over the NPSTMW formation region. The declines of heat loss and southward Ekman transport, owing to the wind stress anomalies, further prohibit the upper-ocean convection and mixed layer deepening and cooling. Additionally, the insufficient development of wintertime MLD in 2018−2021 may also be correlated with the significantly intensified preconditioning of near surface stratification (<150 m depth) due to the persisting near surface warming.

# 1 Introduction

North Pacific Subtropical Mode Water (NPSTMW) is one of the most remarkable low potential vorticity (PV) (i.e., the low stratification) water masses that can be found in the subsurface layer of the western part of the subtropical gyre in the North Pacific Ocean (Oka & Qiu, 2012; Suga et al., 1989). The NPSTMW formation processes are involved in transformation, formation, and subduction of the water mass in upper-ocean (i.e., thermodynamic and dynamic processes, Marshall et al., 1999; Nishikawa et al., 2013). Hence, the mode waters retain the memory of the atmospheric condition at the formation time and are key to understanding interannual-to-interdecadal climate variability (Wu et al., 2020, 2021). The reemergence areas of winter sea surface temperature (SST) anomalies in the Northern and the Southern Hemisphere corresponded to the mode water formation areas (Hanawa and Sugimoto, 2004). The change of water properties from the formation area to the reemergence area, clearly indicate that waters with winter SST anomalies move from the formation area to the reemergence area in a year (Sugimoto and Hanawa, 2005). They play an important role in modulating the ocean ventilation process accompanying biogeochemical variability (Oka et al., 2015, 2019).

NPSTMW are formed in a deep wintertime mixed layer (ML) south of the Kuroshio and the Kuroshio Extension (KE), controlling the ventilation process of the upper-ocean, due to the ocean-atmospheric interaction in wintertime (Oka and Qiu, 2012). Many previous observations and numerical experiments have studied the related factors of interannual to decadal variations in NPSTMW. The early studies have pointed out that the few strong wintertime air-sea heat flux loss events driven by the East Asian monsoon (e.g., Hanawa & Kamada, 2001; Masuzawa, 1969; Rainville et al., 2007; Suga & Hanawa, 1995; Taneda et al., 2000) dominate the interannual variations in NPSTMW formation. Davis et al. (2011) then identified the interannual variability of NPSTMW volume is correlated with the role of the surface heat flux and large-scale wind stress patterns due to the Pacific decadal oscillation (PDO, Newman et al., 2016) shift with a zero-time lag. A negative (positive) PDO index implies a weaker (stronger) Aleutian low (AL), weaker (stronger) westerlies, and higher (lower) sea surface temperature (SST) in the western North Pacific subtropical gyre region (Qiu, 2003). A weaker (stronger) wind stress over the NPSTMW formation region results in less (more) heat loss to the atmosphere and less (more) NPSTMW. In addition, the weaker westerlies in the NPSTMW formation area produces less Ekman transport of the cold water from north and the resulting warmer subsurface ocean inhabits wintertime convection and deep ML formation. However, Qiu & Chen (2006) attributed the decadal variability of NPSTMW to the role of the first-mode baroclinic Rossby waves and quasi-decadal variability of the dynamic state of the KE jet in associated with the PDO. This mechanism has been demonstrated by the successive studies (e.g., Cerovečki & Giglio, 2016; Oka et al., 2015; Qiu et al., 2007; Sugimoto & Hanawa, 2010; Toyama et al., 2015). During the positive (negative) PDO phase, namely when the AL is stronger (weaker), while positive (negative) wind stress curl anomalies that are generated give rise to the enhanced Ekman flux divergence, resulting in negative (positive) sea surface height (SSH) and main thermocline depth (MTD) anomalies in the central North Pacific, and then propagate westward at the speed of the first-mode baroclinic Rossby waves. When the anomalies reach the area east of Japan with a time lag of 3−5 years, KE turns into an unstable (stable) state accompanying a high (low) regional eddy activity (Qiu and

Chen, 2005). Especially, during the unstable KE period, more cyclonic eddies are detached from KE to the southern recirculation gyre (Sasaki and Minobe, 2015), supplying the relatively high PV water from north of KE to hinder the formation of deep ML. Anomalous shoaling of the MTD and stronger background stratification in the NPSTMW formation region, both associated with a negative PDO phase, provide unfavorable oceanic conditions for the development of deep wintertime ML associated with the NPSTMW (Sugimoto and Hanawa, 2010; Sugimoto and Kako, 2016). Cerovečki and Giglio (2016) also demonstrated that the strong NPSTMW density decrease that documented by Argo in 2009 is modulated by a negative density anomaly started to propagated westward from the central Pacific about three years later when the PDO index switched from positive to negative. Finally, the surface density decrease provided oceanic preconditioning for preferential surface formation of a lighter variety of NPSTMW. Cerovečki et al. (2019) expand on the Davis et al. (2011) analysis of the strong NPSTMW volume decrease in 1996−1999, they also found that strong NPSTMW density decrease

played a crucial role in the strong NPSTMW volume decrease in 1999. The near-surface density decrease is caused by temperature increase, in turn caused by an increase in KE geostrophic transport at the time when the KE jet changes from a contracted into an elongated state starting in 1997 (Qiu, 2000). Both processes increased the PV in the NPSTMW region, decreasing the volume of water in the NPSTMW density range. Although locally governed by different physical processes, both NPSTMW volume and density decrease are part of quasi-decadal variability caused by basin-wide changes in wind stress curl.

In August 2017, KE switched from an unstable state to a stable state in association with the occurrence of the Kuroshio large-meander (LM) path south of Japan (Figure 1), although negative SSH and MTD anomalies associated with the positive PDO phase were arriving from the central North Pacific (Qiu et al. 2020). Since then, Kuroshio LM and the stable state of the KE have lasted for more than six years (Qiu and Chen, 2021; Qiu et al., 2023; Usui, 2019), while the NPSTMW volume has declined (Oka et al., 2021). It is worth noting that the current stable KE state seems to have begun with the initiation of LM (Qiu et al., 2020), it has also been supported by basin-wide wind forcing (Qiu et al., 2023). Thus, it is of interest to examine what predominantly determines such variability of NPSTMW (2018−2021). By analyzing in situ observational data,

we are about to identify the physical processes governing the persisting decline of NPSTMW in recent years when the KE jet is stable. Argo observations were used to gain a better understanding of the temporal variability of the NPSTMW, and to identify its associated atmospheric forcing and oceanic dynamics. The rest of this paper is structured as follows: Section 2 describes the data, methods and the mean characteristics of NPSTMW defined in this study. Section 3.1-3.3 describes the recent decreases of NPSTMW volume and density corresponding to the subduction and surface formation rate anomalies, respectively. The mechanisms that give rise to interannual variations in the subduction and formation of NPSTMW is further discussed in Section 3.4-3.5. Section 4 contains a summary and concluding remarks.

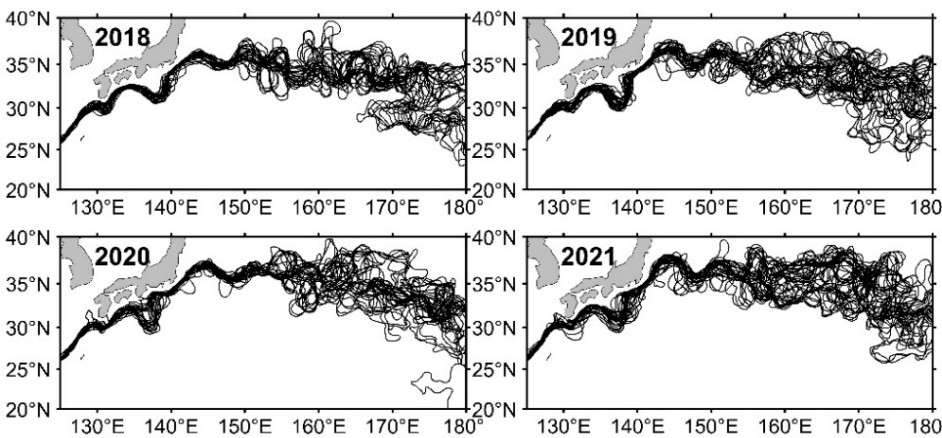

**Figure 1: Yearly (in the Kuroshio LM period) paths of the Kuroshio and its extension defined by 110 cm contours in the weekly SSH fields that reconstructed from Copernicus Marine Environment Monitoring Service satellite-derived altimetry SSH anomaly data and mean dynamic topography data. Here paths are plotted every 14 days.**

## 2 Data and Processing Procedures

### 2.1 Data

The formation and distribution of the NPSTMW are briefly located in an area bounded on 20°N−40°N and on 125°E−180° (east of the islands of Japan, Liu et al., 2017). We use the most recent updated gridded Argo product with a horizontal resolution of 1°×1° obtained by Roemmich and Gilson (referred to as the RGA dataset below; Roemmich & Gilson, 2009), which includes the Argo-only derived monthly temperature and salinity fields covering the period from 2004 to 2021. We also use the other updated gridded dataset (horizontal resolution: 1°×1°, vertical resolution: 25 levels from 10 to 2000 m) provided by JAMSTEC, which includes Argo data and other available water temperature and salinity data monthly since January 2001 (referred to as the MOAA-GPV dataset below; Hosoda et al., 2008). In addition, we also use individual Argo profiles, owing to their better vertical resolution than the gridded data, to ensure the recent variability of NPSTMW in the NPSTMW formation region and enable a detailed description of NPSTMW change. The quality-controlled individual Argo profiles are edited according to the procedures of Oka et al. (2007). The profiles almost cover the whole region of the NPSTMW formation and distribution, especially in the zone west of 155°E (Figure 2a).

The monthly atmospheric variables of net surface heat flux ($Q_{net}$, positive into the sea surface; the sum of latent heat flux, sensible heat flux, net surface longwave radiation flux, and net shortwave radiation flux), evaporation ($E$), precipitation ($P$), sea level pressure (SLP) and wind stress for periods from 2004 to 2021 are from the ECMWF ERA5 reanalysis with a horizontal resolution of 1/4°×1/4° (Hersbach et al., 2019). The zonal and meridional surface geostrophic velocity anomalies, sea surface height (SSH) anomaly data and mean dynamic topography data are provided by Copernicus Marine Environment

Monitoring Service (CMEMS). The CMEMS dataset derived from satellite altimetry has a one-day temporal resolution and a 1/4° spatial resolution from January 1993 to December 2021.

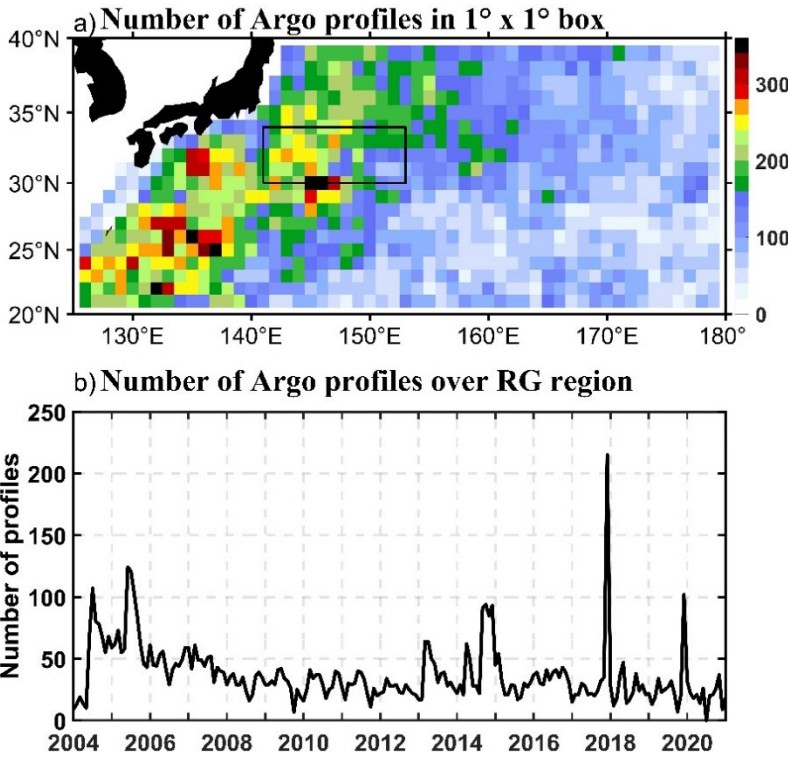

Figure 2: Number of Argo profiles in (a) each 1°×1° box (black solid box shows the recirculation gyre region south of KE), and (b) over the entire the upstream KE recirculation gyre (30°N−34°N, 141°E−153°E) for the period from January 2004 to December 2021 (Note that profiles north of the KE are excluded).

## 2.2 Mean characteristics of the NPSTMW

The NPSTMW is formed in the deep winter (January-March) ML (Suga and Hanawa, 1990). In this study, the NPSTMW is defined as a water mass with a low-PV (PV<$2.0\times10^{-10}$ $m^{-1}$ $s^{-1}$) in the potential density range of $\sigma_\theta$=25.0−25.5 kg $m^{-3}$ (Figure 3). The PV is defined as $PV = -(f / \rho)(\partial\sigma_\theta/\partial z)$, where $f$ is the Coriolis parameter, $\rho$ is a reference density (1025 kg $m^{-3}$), $\sigma_\theta$ is the potential density (Liu et al., 2017). The mixed layer depth (MLD) is defined as the depth at which the potential density is different from the sea surface (10 m) density by 0.125 kg $m^{-3}$ (Levitus, 1982).

The narrow latitudinal extent of the NPSTMW formation extends from 30°N to the KE jet near 35°N (i.e., the upstream KE recirculation gyre; referred to as the RG region below). The part of the NPSTMW then can be advected downstream to the date line and to the subtropical front (Oka & Qiu, 2012; Qiu et al., 2006). Due to focusing on individual profiles detecting NPSTMW formation in the RG region south of the KE (30°N−34°N, 141°E−153°E; Figure 2a), we use only profiles with

temperature (T>16°C) at 200 m to minimize the possibility of identifying subarctic profiles coming from north of the KE, according to the procedures of Sugimoto & Kako (2016). Figure 2b displays the number of profiles in which the MLD and thickness of NPSTMW was detected in the RG region south of KE, revealing that more than 30 profiles are available for most months. Temperature and salinity at each profile are interpolated onto a 1 m vertical grid using the Akima spline (Akima, 1970). We use a 12°C isotherm as an indicator of the MTD in the western part of the North Pacific Subtropical gyre (Uehara et al., 2003).

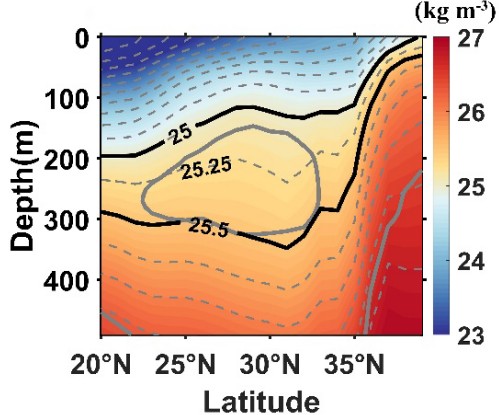

**Figure 3: The meridional section at 150°E of climatological annual mean potential density (color shading and thin gray dashed contours with $\sigma_\theta$=0.25 kg m$^{-3}$ interval) based on the 2004−2021 period averages of the RGA dataset. The thick black solid contours ($\sigma_\theta$=25.0 kg m$^{-3}$ and $\sigma_\theta$=25.5 kg m$^{-3}$) are the isopycnals bounding the density range of NPSTMW, and the thick gray solid contour bounds the low-PV pool with PV<2×10$^{-10}$ m$^{-1}$ s$^{-1}$.**

## 2.3 Formation and subduction of NPSTMW

The subduction rate is defined as the rate at which the water mass is transferred from the ocean surface layer into the main thermocline. The instantaneous subduction rate ($s(t)$) of water mass that leaves the ML in the Eulerian coordinates is given by (Cushman-Roisin, 1987):

$$s(t) = -\left(\frac{\partial h}{\partial t} + \mathbf{u} \cdot \nabla h + w\right), \tag{1}$$

where $h$ is the mixed layer depth (MLD). The rate of mixed layer shoaling is the temporal induction ($-\frac{\partial h}{\partial t}$). The lateral induction rate at which the water is swept beneath the shallowing MLD is by horizontal advection ($-\mathbf{u} \cdot \nabla h$), -$w$ refers to the vertical pumping and $\nabla$ is the horizontal gradient operator. Using the temperature and salinity profiles at each grid point from the RG dataset, we compute the horizontal geostrophic velocity field $\mathbf{u}$ relative to a reference level of 1000 m (Oka et

al., 2011) with a horizontal resolution of $1° \times 1°$. The vertical pumping $w = \mathrm{curl}(\boldsymbol{\tau}/(\rho f))$ is calculated from wind stress ($\boldsymbol{\tau}$) provided by the ERA5, and $\rho = 1025$ kg m$^{-3}$.

Practically, the formation process of mode water is that the low-PV water mass at the base of late winter ML is subducted into the ocean interior (Oka & Qiu, 2012). Thus, the PV of subducted water is related to the subduction rate by (Williams, 1989, 1991):

$$\mathrm{pv} = \frac{f}{\rho} \frac{\frac{\partial \rho_{\mathrm{m}}}{\partial t} + \mathbf{u}_{\mathrm{b}} \cdot \nabla \rho_{\mathrm{m}}}{s}, \tag{2}$$

where $\mathbf{u}_{\mathrm{b}}$ is the horizontal velocity at the base of the mixed layer. $\rho_{\mathrm{m}}$ and $\rho$ are the mixed layer density and reference density, respectively. $s$ respresents the subduction rate.

Since the instantaneous subduction rate fluctuates considerably, the annual mean subduction rate ($s_{\mathrm{ann}}$) is defined as (Wu et al., 2021):

$$s_{\mathrm{ann}} = \frac{1}{T_0} \int_{T_{\mathrm{s}}^{\mathrm{ef}}}^{T_{\mathrm{e}}^{\mathrm{ef}}} s(t) dt, \tag{3}$$

where $T_0$ is one year, $T_{\mathrm{s}}^{\mathrm{ef}}$ and $T_{\mathrm{e}}^{\mathrm{ef}}$ are the times when effective subduction starts and ends, respectively. The effective subduction period is fixed in late winter (February-April, Oka et al., 2015).

Before the effective subduction period in late winter, the fluid passes laterally across outcropping isopycnals at sea surface in winter; the transformation of fluid that makes up a particular density layer is modified either by surface buoyancy flux, vertical diffusion, or eddy buoyancy fluxes (e.g.,Garrett & Tandon, 1997; D. Marshall, 1997; J. Marshall et al., 1999; Walin, 1982). In practice, the water mass transformation ($F$) at time ($t$) is calculated using discrete density intervals but considering the air-sea flux only (neglecting interior mixing, Cerovečki et al., 2019; Cerovečki & Marshall, 2008):

$$F(\sigma_\theta, t) = -\frac{\partial}{\partial \sigma_\theta} \iint_{A_{\sigma_\theta}} B_{\mathrm{surf}} dA, \tag{4}$$

Here, $A_{\sigma_\theta}$ is the surface area of the outcrop window for $\sigma_\theta$ within the density interval $[\sigma_\theta - \Delta\sigma_\theta/2, \sigma_\theta + \Delta\sigma_\theta/2]$ separated by the density increment $\Delta\sigma_\theta$. Transformation has units of Sverdrups (Sv; 1Sv=10$^6$ m$^3$ s$^{-1}$). The surface buoyancy flux ($B_{\mathrm{surf}}$) is given by:

$$B_{\mathrm{surf}} = -\frac{\alpha}{C_p} Q_{\mathrm{net}} + \beta\rho_0 S_0 (E - P), \tag{5}$$

Where $\alpha$ and $\beta$ are the thermal expansion and saline contraction coefficients, respectively. $C_p$ is the specific heat of seawater. $S_0$ is the sea surface water salinity. $\rho_0$ is the sea surface density.

The convergence of the air-sea transformation rate ($F$) in the density interval $[\sigma_\theta - \Delta\sigma_\theta/2, \sigma_\theta + \Delta\sigma_\theta/2]$ yields the air-sea formation rate (Guo et al., 2018; Small et al., 2022):

$$M(\sigma_\theta, t)\Delta\sigma_\theta = F\left(\sigma_\theta - \frac{\Delta\sigma_\theta}{2}, t\right) - F\left(\sigma_\theta + \frac{\Delta\sigma_\theta}{2}, t\right), \tag{6}$$

Here, $M$ is the water mass accumulation per unit density (i.e., the convergence of water mass transformation in density space) and $M\Delta\sigma_\theta$ is referred to as the water mass formation (with units of Sv).

## 2.4 ML heat budget analysis

NPSTMW core, defined as the PV minima within the low-PV layer, is considered to preserve the properties of the deep ML from which it subducts (Oka et al., 2011; Qiu, 2002). The potential temperature at the NPSTMW core is the core layer potential temperature (CLT). In order to assess quantitatively the effect of the interannual variability of the Kuroshio and its Extension on the mixed layer temperature (MLT), we perform the heat budget analysis below in the surface ocean ML (Qiu & Kelly, 1993; Sugimoto & Kako, 2016):

$$\frac{\partial T_m}{\partial t} = \frac{Q_{net} - q_d}{\rho C_p h} - \frac{w_e}{h}(T_m - T_d) - \mathbf{u}_e \cdot \nabla T_m - \mathbf{u}_g \cdot \nabla T_m, \tag{7}$$

Where $T_m$ denotes the MLT, $Q_{net}$ is the net surface heat flux, $\rho$ is the reference density of seawater, $h$ is the MLD, $\mathbf{u}_e$ is the Ekman velocity and is related to the surface wind stress vector $\boldsymbol{\tau}$ by $\boldsymbol{\tau} \times \mathbf{k}/(\rho f h)$, $w_e$ is the entrainment velocity, $T_d$ is the water temperature at 20 m below the base of the ML, $\mathbf{u}_g$ and is the sea surface geostrophic velocity. Here $q_d = q(z)$ is the downward radiative flux at the base of ML, and is given by

$$q(z) = q(0)\left[R\exp\frac{-z}{\gamma_1} + (1-R)\exp\frac{-z}{\gamma_2}\right], \tag{7-1}$$

Where $q(0)$ is the surface downward radiative flux, $R$ (=0.77) is the separation constant, and $\gamma_1$ (=1.5 m) and $\gamma_2$ (=14 m) are the attenuation length scales. We estimate entrainment velocity diagnostically based on the MLD following the method of Qiu (2000):

$$w_e = \begin{cases} \Delta h/\Delta t, & \Delta h/\Delta t > 0 \\ 0, & \text{otherwise} \end{cases}, \tag{7-2}$$

Herein, we refer to the five terms in Eq. (7) as the temperature tendency term, the air-sea heat exchange term, the vertical entrainment term, the Ekman advection term, and the geostrophic advection term, respectively.

## 2.5 Steric height change

Outside the tropics, the SSH signal derived from satellite altimetry (CMEMS dataset) is dominated by large-scale seasonally varying steric processes. Steric height changes $\eta'_s$ are caused by the expansion or contraction of the water column (Stammer

D.1997). It can be written to show the contribution from heat and salt as (Vivier et al., 1999)

$$\eta'_s = \int_{-H}^{0} \alpha T' dz - \int_{-H}^{0} \beta S' dz , \qquad (8)$$

The integration is limited to the upper-ocean that is affected by atmospheric buoyancy fluxes. Where $T'$ and $S'$ are the departures from temporal mean temperature and salinity, whereas the first term of the right-hand side of Eq.(8) is substantially larger than the second term in most of the Pacific Ocean. Thus, only thermal processes are considered. As the

steric height signals are not the focus of this study, $\eta'_s$ due to thermal expansion is removed hereinafter in the SSH signal derived from satellite altimetry. Removal of the steric height signals has little impact upon the SSH signals that are indirectly induced by the surface buoyancy fluxes (Qiu, 2000).

## 3 Results

### 3.1 NPSTMW volume loss and density decrease in 2018-2021

The monthly time series of NPSTMW volume and that binned into potential density at $\sigma_\theta = 0.05$ kg m$^{-3}$ are shown in Figure 4a and 4b. The NPSTMW volume change is modulated by the variability of KE state since 2004 (Oka et al., 2021). In a short time of 2006−2009 when the KE jet is unstable, the NPSTMW volume has a dramatic decrease. The volume of NPSTMW has restored and subsequently increased largely during 2012−2015 when the KE is stable. But in recent stable KE period, it is worthy to note that NPSTMW volume in analysis region gradually decreased since 2018. The mean total volume of

NPSTMW drop by ~21% during 2018−2021 ($7.38 \times 10^{14}$ m$^3$) on average relative to 2012−2015 ($9.29 \times 10^{14}$ m$^3$) (Figure 4a). Oka et al. (2021) investigated NPSTMW change in the region (15ºN–40ºN, 120ºE–170ºW) using Argo floats observations. They showed that annual-mean volume increased lightly in the northeast region (28ºN−40ºN, 140ºE−170ºW) from 2017 to 2019, while that in the other region decreased significantly. In fact, the NPSTMW in the whole region have gradually decreased in recent years. Hence, the recent decline of NPSTMW observed in RGA dataset is definitely consistent with the

results of Oka et al. (2021). Particularly, the mean NPSTMW volume in denser density range (approximately $\sigma_\theta > 25.2$ kg m$^{-3}$) appears to decrease from $11.07 \times 10^{13}$ m$^3$ in 2012−2015 to $8.77 \times 10^{13}$ m$^3$ in 2018−2021 (Figure 4b). Meanwhile, we also conduct the Mann-Kendall test on the NPSTMW volume in denser density range (approximately $\sigma_\theta > 25.2$ kg m$^{-3}$) from

2012 to 2018 following the method of Xu et al. (2022). For the present study, the null hypothesis of no trend is rejected when the absolute value of the standard deviation $|Z| > 1.96$ at the significance level is 0.05. There is a decreasing trend for the change of the NPSTMW volume in denser density range ($Z = -3.09$). It suggests the density of NPSTMW has declined accompanied with the NPSTMW volume loss in 2018−2021. The averaged maximum NPSTMW thickness in April to May when annual newly NPSTMW formed also started to gradually decrease since 2018. The mean thickness of NPSTMW in April to May from 2018−2021 has approximately dropped by 50 m. There similarly seems to be a persistent loss of newly NPSTMW formation in 2018−2021. To examine the results of the gridded RGA dataset and elongated the timescale of RGA dataset, we also analyze the results from the MOAA-GPV dataset. Besides 2012−2015, the KE is also in the stable state during 2003−2005. The volume and density changes of NPSTMW in 2003−2005 has the same pattern with that in 2012-2015 (Figure 5). The variability of KE dynamic state is the significant mechanism in modulating the variabilities of NPSTMW in both periods of stable KE state above pointed out by previous studies (Oka et al., 2015; 2019). In addition, the increase of NPSTMW volume and density is dramatic in 2012−2015 compared to that in 2003−2005. Thus, the comparisons of changes of NPSTMW and mechanisms between in 2012−2015 and 2018−2021 (both in relatively long time of stable KE period) is mainly presented hereinafter.

The NPSTMW volume and density increase in 2012−2015 coincided with a period when the KE is in its stable dynamic state. However, what causes the anomalous decline of the NPSTMW during a recent period of stable KE state? Was the observed decrease of newly formed NPSTMW volume and density in 2018−2021 predominantly caused by a decrease in surface formation in the NPSTMW density range and a decrease in subduction of low-PV water in the NPSTMW density range? Furthermore, was the observed decrease of NPSTMW in a recent stable KE period related in some ways to local and distant effects of atmospheric forcing or oceanic precondition, as suggested by Davis et al. (2011) and Sugimoto & Hanawa (2010)? We address these questions in the subsequent sections.

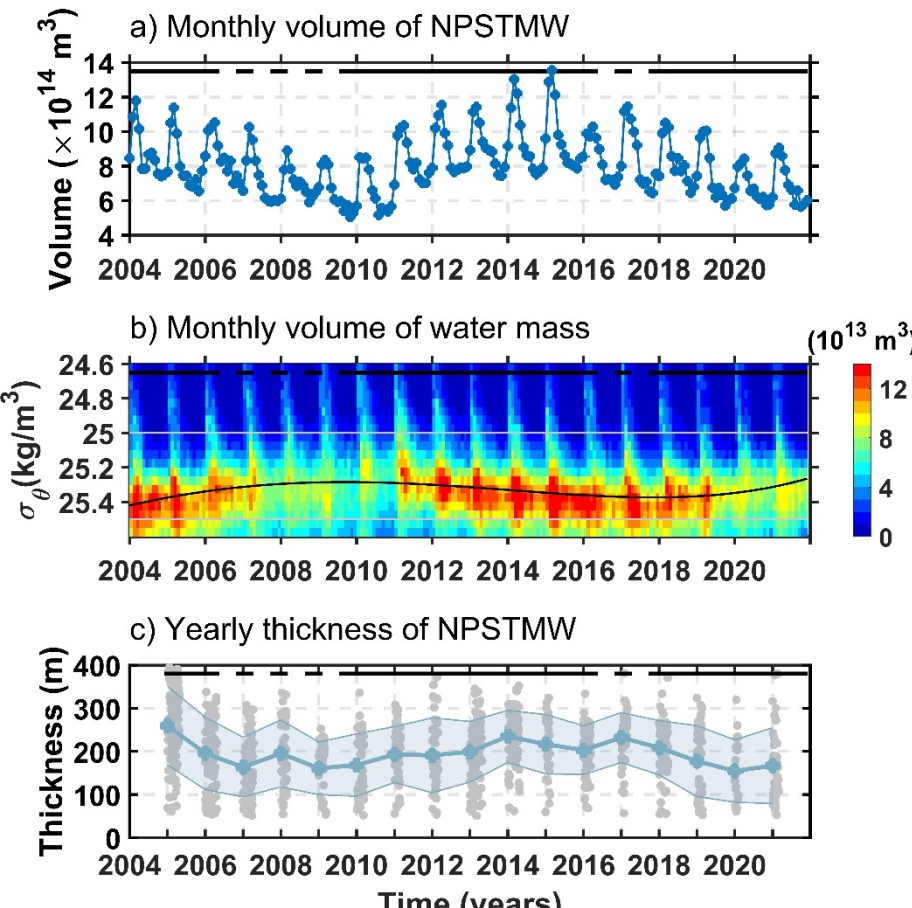

Figure 4: (a) Monthly time series of volume of NPSTMW with $\sigma_\theta$=25.0−25.5 kg m$^{-3}$ obtained from the RGA dataset. (b) Monthly averaged water mass volume (10$^{13}$ m$^3$) estimated from the RGA dataset for the years 2004−2021, in the potential density range $\sigma_\theta$=24.6−25.6 kg m$^{-3}$ (color, right bar) with PV<2×10$^{-10}$ m$^{-1}$ s$^{-1}$. The black line is cubic fit to monthly volume maximum. Gray contours of the 25 kg m$^{-3}$ and 25.5 kg m$^{-3}$ isopycnal surfaces are constant when plotted against density. Both estimates are obtained over the analysis region 20°N−40°N, 125°E−180°, east of Japan. (c) The time series of NPSTMW thickness during the period of yearly maximum thickness (April−May). Gray circles represent values observed from individual Argo profiles in the RG region. The circles-line represents mean values and the shading around each line shows the standard deviation. Solid (dashed) bars indicate stable (unstable) periods of the KE.

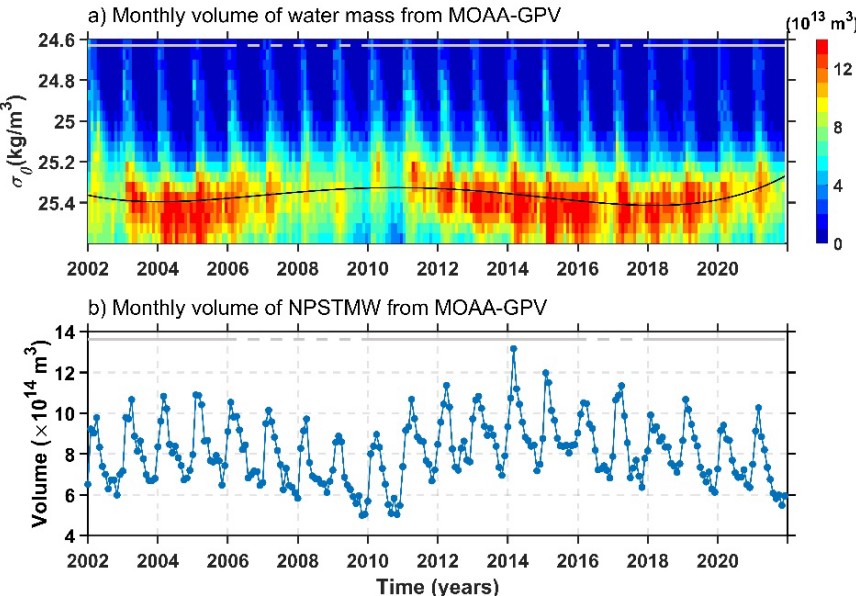

**Figure 5: (a)** Monthly averaged water mass volume ($10^{13}$ m³) estimated from the MOAA-GPV dataset for the years 2004−2021, in the potential density range $\sigma_\theta$=24.6−25.6 kg m⁻³ (color, right bar) with PV<2×10⁻¹⁰ m⁻¹ s⁻¹. The black line is cubic fit to monthly volume maximum. **(b)** Monthly time series of volume of NPSTMW with $\sigma_\theta$=25.0−25.5 kg m⁻³ obtained from the MOAA-GPV dataset. Both estimates are obtained over the analysis region 20°N−40°N, 125°E−180°, east of Japan. Solid (dashed) bars indicate stable (unstable) periods of the KE.

### 3.2 Decrease in the NPSTMW surface formation

Surface water mass transformation and formation rates have been introduced in section 2.3. Inputs to this analysis are monthly estimates of air-sea buoyancy flux and of surface density in winter. First, maps of annually averaged surface formation rate in the NPSTMW density range during 2012−2015 and 2018−2021 when the KE is similarly in a stable state are shown in Figure 6. The averaged areas of the NPSTMW outcrop window (i.e., the averaged formation area of the NPSTMW) of 2.67×10¹² m² during 2018−2021 do not shrink remarkably relative to 3.08×10¹² m² during 2012-2015 (Figure 6). Because the less changes in the position of KE jet when KE is in a stable state period, not as the case in 2006−2009 when the smallest NPSTMW outcropping window was attributed to the spatially convoluted path of the KE in the unstable state (Cerovečki & Giglio, 2016). In a stable KE period of 2012−2015, the strong surface formation tends to happen in the upstream of KE (141°E-153°E). However, the averaged surface formation in the upstream of KE has declined in recent stable KE period (Figure 6). Equation (4) shows that the water mass transformation will be weak owing to the less surface ocean buoyancy loss over the outcropping window.

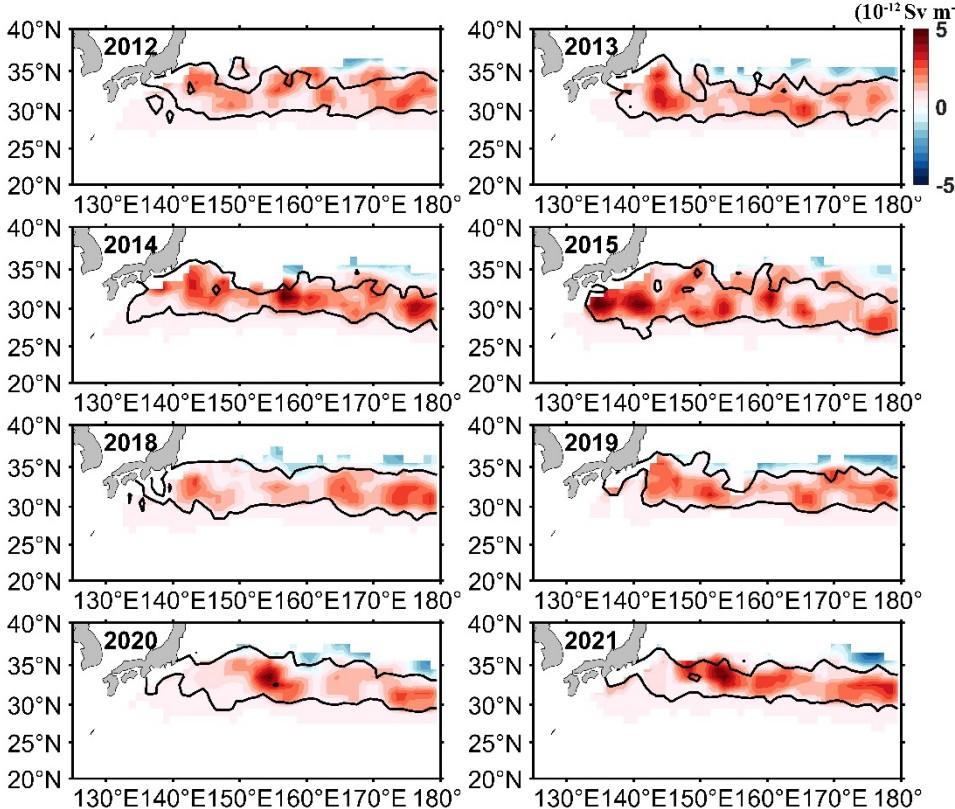

**Figure 6: Maps of averaged surface formation rate per unit area ($10^{-12}$ Sv m$^{-2}$) during wintertime in the NPSTMW density range. The solid lines bound the $\sigma_\theta$ =25.0−25.5 kg m$^{-3}$ outcrop region time averaged from January to March.**

Furthermore, in each of years considered, the annually averaged surface transformation and formation rates binned into the potential density at $\sigma_\theta$=0.1 kg m$^{-3}$. Figure 7 (a) and (b) show the averaged surface transformation and formation rate in 2012−2015 and 2018−2021, respectively. The averaged surface formation rate over the time period 2018−2021 increase largely at the $\sigma_\theta$=25.1 kg m$^{-3}$, while the negative formation rate occurs in 2012−2015. It suggests that the NPSTMW in the lighter NPSTMW density range (approximately $\sigma_\theta$<25.2 kg m$^{-3}$) accelerate to form in 2018−2021. Meanwhile, over the time period of 2018−2021, the peak formation rate shift toward lower densities (approximately $\sigma_\theta$<25.2 kg m$^{-3}$) compared to time period of 2012−2015, so that in 2018−2021 only the lighter variety of water in the NPSTMW density range is replenished by surface formation. Over the time period 2018−2021, averaged transformation rates (15.22 Sv) in the lighter NPSTMW density range (approximately $\sigma_\theta$<25.2 kg m$^{-3}$) compared to that in the time period 2012−2015 (19.26 Sv) were greatly reduced by 21%. Decreased transformation as well reflects decrease of surface buoyancy loss in 2018−2021, as indicated by Eq. (4).

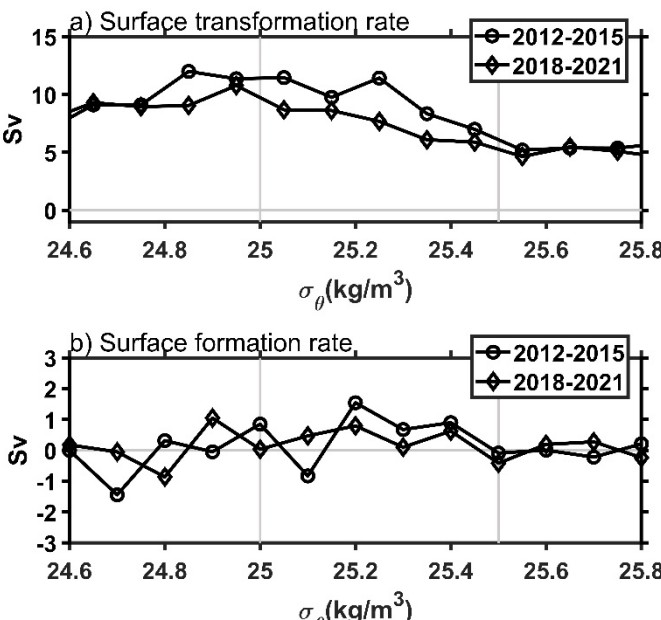

Figure 7: (a) The averaged surface transformation rate and (b) formation rate during wintertime of 2012−2015 and 2018−2021 obtained from ERA5 air-sea buoyancy flux using the Walin (1982) framework within the wintertime isopycnal outcrop region.

**3.3 Decrease in subduction rate and NPSTMW subduction variability**

After formation of the low-PV water within the NPSTMW density range in winter, the newly wintertime low-PV water mass is subsequently subducted into the ocean interior forming NPSTMW in late winter and early spring. Figure 8 compares the patterns of subduction rate between 2012−2015 and 2018−2021. The large subduction of NPSTMW usually occurs in the RG region south of KE, as depicted in Figure 8a. However, the total subduction rate of 2018-2021 reduced significantly in the RG region, even dropped by reaching 90 m year[-1] (Figure 8c). Here, the subduction rate changes in 2018-2021 are predominantly caused by the temporal induction and lateral induction terms, while the vertical pumping term is negligible (Figure 8d−8f). The evident negative anomalies of temporal induction and lateral induction can be attributed to the anomalously shallow MLD located at the south of KE (Figure 9). The MLD in February−March in the RG region from individual Argo float profiles (Figure 9d) presents the same way as Figures 9c: the MLD in 2018-2021 is anomalously shallower than 2012-2015 by approximately 50 m on average in a stable KE period. Additionally, it is interesting to note that the mean outcropping isopycnals in the NPSTMW density range has a clear northward movement in 2018−2021, comparing to that located in 2012−2015. Wu et al. (2021) and Kawakami et al. (2023) have revealed that the northward KE shift is mostly caused by the trend of wind stress curl in the North Pacific during 1993−2021. The surface warming due to the northward KE shift may cause the northward movement of the outcropping isopycnals and inhibit the deepening of the ML south of KE in the winter of 2018−2021.

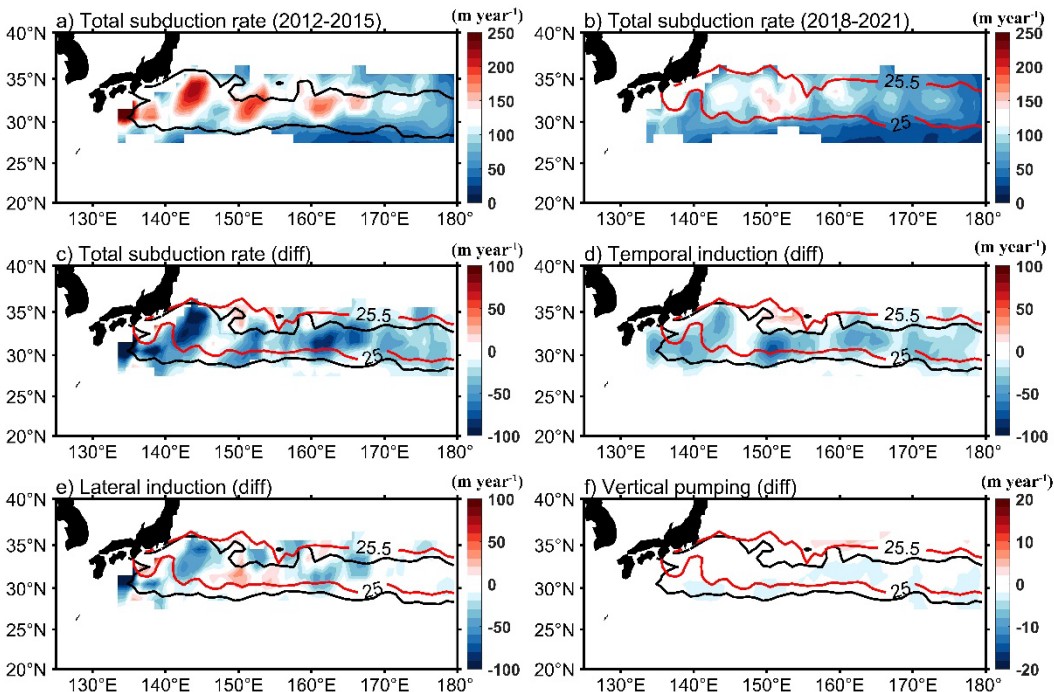

Figure 8: (a) the total subduction rate of 2012−2015 (b) the total subduction rate of 2018−2021 (c) the difference (the values of 2018−2021 minus that of 2012−2015) between the subduction rate (m year⁻¹) and three components responsible for the subduction rate change in the outcrop region: (d) temporal induction ($-\frac{1}{T_0}\int_{T_s^{ef}}^{T_e^{ef}}\frac{\partial \bar{h}}{\partial t}dt$), (e) lateral induction ($-\frac{1}{T_0}\int_{T_s^{ef}}^{T_e^{ef}}\left(\bar{\mathbf{u}}\cdot\nabla\bar{h}\right)dt$), and (f) vertical pumping ($-\frac{1}{T_0}\int_{T_s^{ef}}^{T_e^{ef}}\bar{w}dt$). Here, $t$ is time, $T_0$ is one year and $T_s^{ef}$ and $T_e^{ef}$ are the times when effective subduction starts and ends, respectively (i.e., from February to April) (Marshall et al., 1993). The outcropping lines of $\sigma_\theta$=25.0 kg m⁻³ (the southernmost one) and 25.5 kg m⁻³ (the northernmost one) are superimposed in the black solid (2012−2015) and red solid contours (2018−2021). The results obtained from RGA/ERA5 datasets with imposing the PV constraint (<2 × 10⁻¹⁰ m⁻¹ s⁻¹ of Eq. (2)).

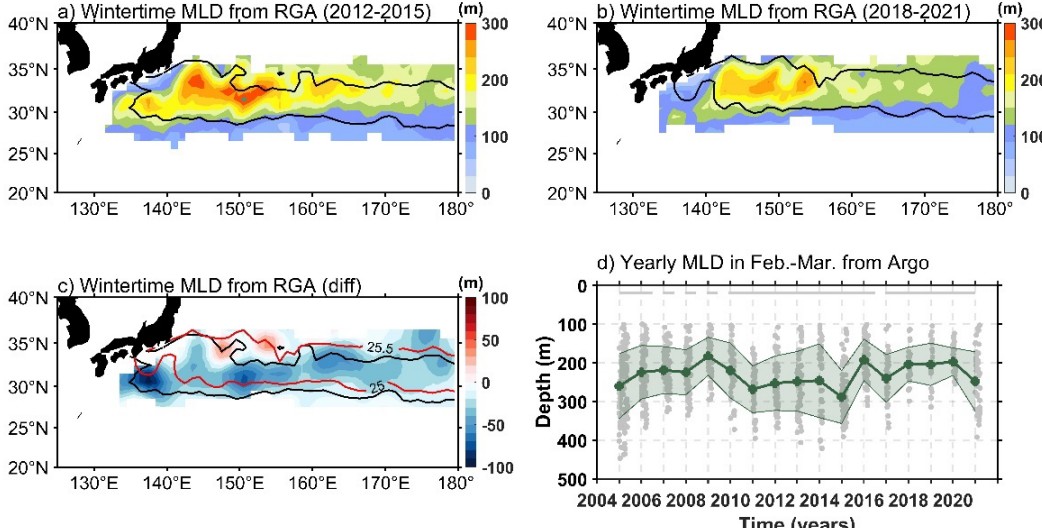

**Figure 9: The wintertime MLD (m) in the outcrop region from the monthly mean RGA dataset in the (a) 2018-2021 and that (b) in 2012−2015. (c) The difference (the values of 2018−2021 minus that of 2012−2015) between (b) and (a). The outcropping lines of $\sigma_\theta$=25.0 kg m$^{-3}$ (the southernmost one) and 25.5 kg m$^{-3}$ (the northernmost one) are superimposed in the black solid (2012−2015) and red solid contours (2018−2021) in the Figure (c). (d) The time series of MLD in February−March in the RG region south of KE (30°N−34°N, 141°E−153°E). Gray circles represent values observed from individual Argo profiles in the RG region. The circles-line represents mean values and the shading around each line shows the standard deviation. Solid (dashed) bars indicate stable (unstable) periods of the KE.**

The tendency of temporal induction and lateral induction both due to the MLD in late winter are consistent with that of the total subduction (Figure 10). The interannual variations in MLD in February to March (observed from individual Argo profiles; Figure 9d) show the similar trend with the total subduction. Meanwhile, the spatial distribution of temporal induction anomalies between 2012−2015 and 2018−2021 (Figure 8d) is also consistent with the wintertime MLD difference in Figure 9c. In 2018-2021, the reduction in temporal induction and lateral induction, representing the effects of horizontal advection and horizontal gradient of MLD in late winter, contribute most of the subduction weakening in the NPSTMW formation region. Thus, the weakening of subduction rate in the NPSTMW formation region is mainly related to changes in local MLD in late winter, which is consistent with previous studies that suggested changes of the subtropical subduction rate in the North Pacific are mainly due to changes of the MLD variation in late winter (Qu and Chen, 2009; Hu et al., 2011; Wang et al., 2015). In 2018−2021, the decrease of NPSTMW subduction in the south of KE region is mostly attributed to the lateral induction and temporal induction volume. Due to the anomalously shallow wintertime MLD, the subduction rate that occurred in the NPSTMW formation region dropped largely in 2018−2021. Consequently, the subduction volume of NPSTMW drop from $5.60\times10^{14}$ m$^3$ in 2015 to $2.99\times10^{14}$ m$^3$ in 2021 (Figure 10). Especially in 2018−2021, the total subduction volume of NPSTMW in the whole analysis region has declined by 37% on average relative to the values of 2012−2015.

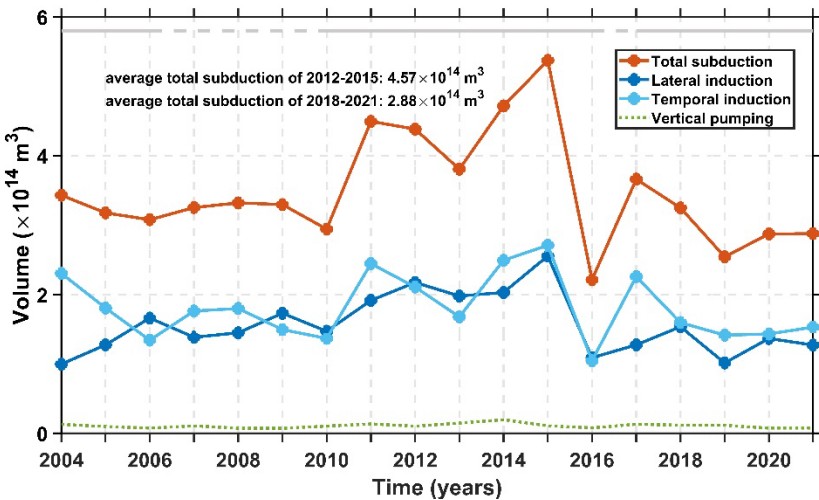

**Figure 10: Annually-averaged total subduction volume and its three components: temporal induction volume ($-\int\limits_{T_s^{ef}}^{T_e^{ef}}\left[\sum\left(\dfrac{\partial \bar{h}}{\partial t}\right)\Delta A\right]dt$)**

**from the monthly mean RGA dataset, lateral induction volume ($-\int\limits_{T_s^{ef}}^{T_e^{ef}}\left[\sum\left(\bar{\mathbf{u}}\cdot\nabla\bar{h}\right)\Delta A\right]dt$) from the monthly mean RGA dataset, and**

**vertical pumping ($-\int\limits_{T_s^{ef}}^{T_e^{ef}}\left[\sum\left(\bar{w}\right)\Delta A\right]dt$) from the monthly mean ERA5 data and annually-averaged volume of NPSTMW over the**

**Northwestern Pacific Ocean (125°E−180°, 20°N−40°N) from the monthly mean RGA dataset for 2004−2021. The results obtained with imposing the PV constraint (<2×10⁻¹⁰ m⁻¹ s⁻¹ of Eq. (2)). Solid (dashed) bars indicate stable (unstable) periods of the KE.**

## 3.4 Interannual variations in CLT and ML heat budget

In order to investigate the interannual variations of NPSTMW properties, the interannual variations in CLT, retaining the wintertime ML water temperature, was depicted in Figure 11. The higher CLT (Figure 11) is significantly consistent with the decrease of NPSTMW volume and density such as 2006−2009 and 2018−2021, which implies that higher MLT and shallower MLD (Figure 9d) in the NPSTMW formation time. On the contrary, the lower CLT in 2012−2015 (Figure 11) is corresponding to the increase of NPSTMW volume and density due to the lower MLT and deeper MLD (Figure 9d) in the NPSTMW formation time. In the previous section, we have qualitatively inferred that the air-sea exchange significantly affects the NPSTMW change. To access directly what degree the time-varying MLT in NPSTMW formation time is controlled by the cumulative heat flux in the upper ocean, we perform an ML heat budget analysis below. Additionally, the strengthening of KE jet in a persisting stable KE period that favors a northward shift of the KE jet may cause warming of the sea surface temperature in the KE region (Qiu, 2000). Isn't this heat advection the important cause of the less formation of NPSTMW after 2018? In order to assess quantitatively the effect of the interannual variability of the Kuroshio and its Extension on the MLT, we diagnose the interannual variability of the cumulative surface heat flux in the cooling season (October of the previous year to March) as following.

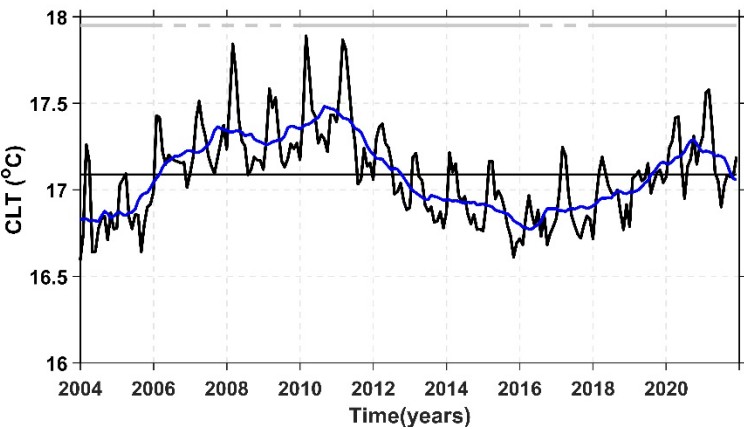

**Figure 11: The black line is the monthly variation of CLT of NPSTMW from RGA dataset in the NPSTMW formation and distribution region (125°E − 180°, 20°N − 40°N), the blue line is smoothed by a 13-month running mean. The black line is climatological CLT of NPSTMW. Solid (dashed) bars indicate stable (unstable) periods of the KE.**

As shown in Figure 12a, the annually averaged net flux anomaly in the cooling season is comparable to the annual

temperature tendency anomaly over the ventilation region south of the KE (141°E−180°, 30°N−34°N). In the cooling season, the sum of the air-sea heat exchange term, the vertical entrainment term, the Ekman advection term, and the geostrophic advection term in the right-hand of Eq.(7) is balanced with the temperature tendency term in the left-hand of Eq.(7) in the wintertime ventilation region of 141°E−180°, 30°N−34°N (Figure 12). The residual term usually includes effects such as eddy heat flux and horizontal diffusion (Qiu & Kelly, 1993). The temperature tendency in climatological average is

counterbalanced by the air-sea heat exchange (41.2%), the vertical entrainment through the base of the ML (26.7%), the Ekman advection (8.2%), and the geostrophic advection (23.9%). Thus, the temporal variability of MLT in the cooling season is dominated by the anomalies of the air-sea heat exchange, the vertical entrainment and the geostrophic advection in climatological average. The Ekman advection makes a small contribution to the local heat balance change. This result is basically consistent with that pointed out by Qiu & Kelly (1993).

However, the average of temperature tendency anomaly during the cooling season of 2018-2021 (Figure 12b) is positive. It is contributed by the air-sea heat exchange (38.0%), the vertical entrainment through the base of the ML (37.0%), the Ekman advection (17.6%), and the geostrophic advection (7.4%). This result demonstrates that, in the NPSTMW formation region, the weak processes of the air-sea heat exchange and the vertical entrainment play an important role in the ML warming during 2018−2021. This quantitative result of weakened air-sea heat loss in ML heat budget also demonstrates that the air-

sea buoyancy flux loss is weak during 2018-2021 suggested in the section 3.2. The weak temperature advection by the Ekman flow also makes contributions to the warming of local MLT. During the time of 2018−2021, the drop of surface buoyancy flux and vertical entrainment lead to the warming in upper-ocean layers (Figure 12). In addition, the Ekman advection also accelerates the regional warming. Warming in the local upper ocean may cause a shoaling in late winter MLD at the same time, especially in the NPSTMW subduction region, which will prevent water subducted into the main

thermocline from the bottom of the mixed layer in late winter (Figures 8 and 10). Thus, the decline of the air-sea heat exchange and the vertical entrainment in the NPSTMW formation region dominantly results in the shoaling of local mixed layer in winter, which in turn leads to local subduction weakening during 2018−2021.

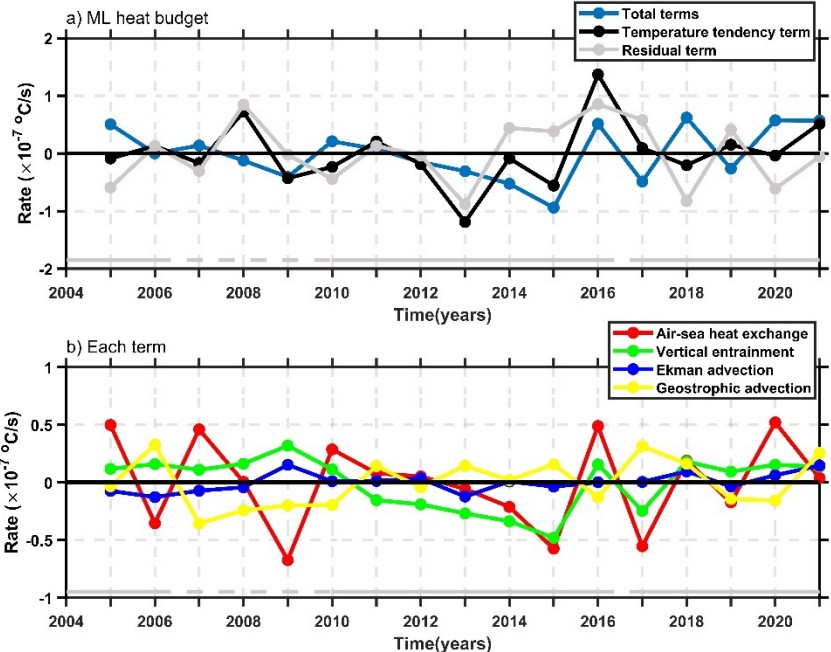

**Figure 12: Yearly time series of the anomaly of values (positive values indicate that the term increases MLT) relative to the 2004−**
**2021 climatology from October of the previous year to March (cooling season) in Eq. (7) for (a) temperature tendency term and residual term, (a) total terms are represented as the sum of (b) the air-sea heat exchange term, (b) the vertical entrainment term, (b) the Ekman advection term, and (b) the geostrophic advection term. Values are averaged in the wintertime ventilation region of 141°E−180°, 30°N−34°N. Solid (dashed) bars indicate stable (unstable) periods of the KE.**

## 3.5 Local and distant effects on NPSTMW properties

The air-sea heat exchange and the vertical entrainment are both associated with the overlying atmosphere. In addition to the KE dynamic state change, the NPSTMW interannual variability has been closely tied to the basin-wide changes of both atmospheric forcing and oceanic precondition associated with the PDO, both in observations (Qiu and Chen, 2006) and numerical model results (Davis et al., 2011). The NPSTMW volume variations in 2006−2009 (2012−2015) (Figure 4a) are closely controlled by the MTD because the PDO-related stronger (weaker) subsurface stratification from the seasonal
thermocline to the main thermocline propagated from the central North Pacific (Figure 13). Meanwhile, Figure 13 also shows the distant effects of MTD (i.e., subsurface stratification) change on NPSTMW formation in the western North Pacific with some time lag (about 3−4 years). Although PDO is positive during 2014−2017, such signals of negative MTD, which propagates westward as oceanic Rossby waves, vanishingly affect the NPSTMW formation region due to the persisting strong sea surface warming (with positive SSH anomaly as its proxy) in NPSTMW formation region during 2018−2021

(Figure 13). However, since 2018, the PDO has transitioned from a positive to negative phase (Figure 13a), with the concomitant positive SSH anomaly and deepened MTD (Figures 13b and 13c). The positive MTD anomaly which weakens the background stratification should have facilitated the development of wintertime MLD and NPSTMW formation, as suggested as that in 2012−2015. On the contrary, the MLD and the subduction of NPSTMW do not have such sufficient developments in the late winter of 2018−2021 (Figures 8 and 9). Additionally, the recently anomalous NPSTMW decline

also follows the PDO phase shift. Thus, the subtle time lag between PDO shift and NPSTMW may have more important implications to investigate next.

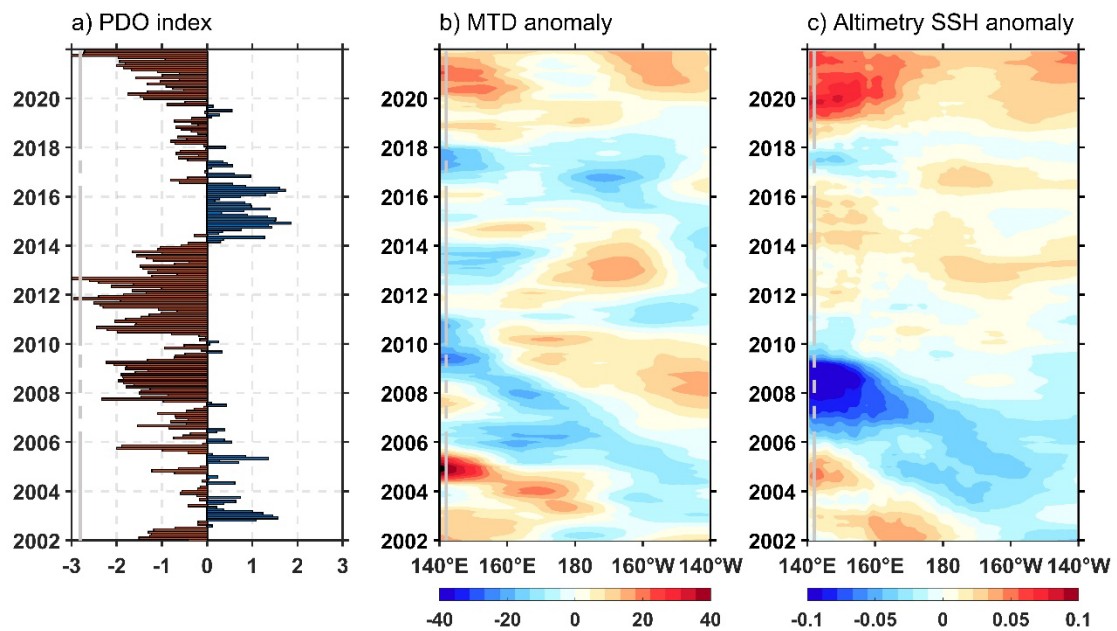

**Figure 13: (a) Monthly values of the PDO index (Mantua, 1999). (b) Longitude–time diagram of MTD anomaly (m) (MTD anomaly obtained from MOAA-GPV dataset) averaged for a zonal band of 30°N−34°N. Positive (negative) values represent deep**

**(shallow) anomalies. (c) as in (b), but for the SSH anomaly (m) (steric height changes are removed). Solid (dashed) bars indicate stable (unstable) periods of the KE.**

    Here, the dynamics and thermodynamics of anomalous NPSTMW variability in 2018−2021, particularly in recently persisting stable KE state, are sought by examining wind stress and surface heat flux associated with an instantaneous response to events in the central Pacific (i.e., the PDO phase shift). The current stable KE state seems to have begun with the

initiation of LM (Qiu et al., 2020), it has also been supported by basin-wide wind forcing (Qiu et al., 2023). The dominant surface wind stress forcing in the midlatitude North Pacific basin is related to the PDO phases (Qiu et al., 2023). When the PDO phase turned from a positive into negative phase in 2018−2021, the AL shifted northward (Figures 14a and 14b) and negative wind stress curl anomalies were generated in the central Pacific (Figure 14c). The basin-scale atmospheric forcing impacts a remarkable influence on the subtropical gyre region (Figure 15). It appears that the AL northward shift (Figure 14)

and subsequent changes of the intensity of westerlies corresponds to the reduced NPSTMW formation in 2018-2021

compared with that in 2012−2015. The weaker westerlies in NPSTMW formation area leads to less Ekman transport of the cold water from the north (Figure 15). The averaged magnitude of southward meridional Ekman transport ($T_y = -\tau_x/(\rho f)$) occurred in the NPSTMW formation region has declined by 31% during 2018−2021 relative to 2012−2015. The horizontal heat advection rate by the Ekman velocity ($-\left(\mathbf{u}_e \cdot \nabla T_m\right)$) has correspondingly declined by 16% during 2018−2021 relative to 2012−2015. The mean heat flux loss in the outcrop window during 2018-2021 has decreased by 14% relative to 2012−2015 (Figure 15). During the time of 2018−2021, the weak surface heat flux loss may be associated with the weak regional wind stress in winter (Figure 15).

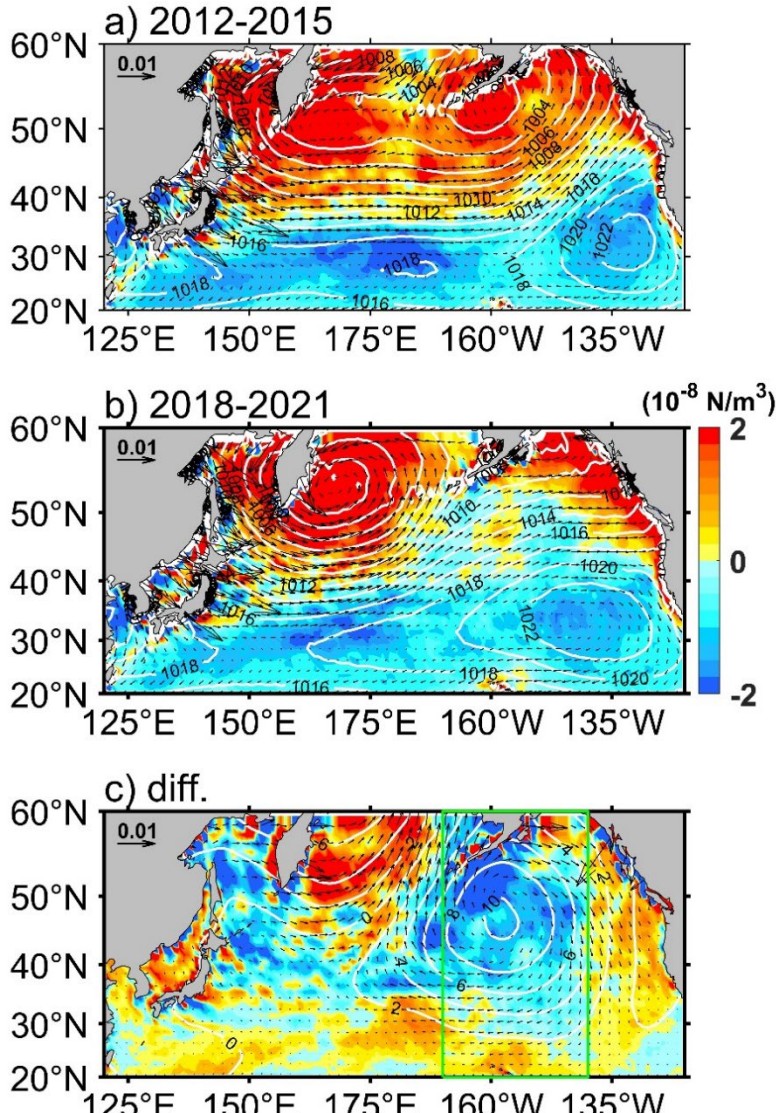

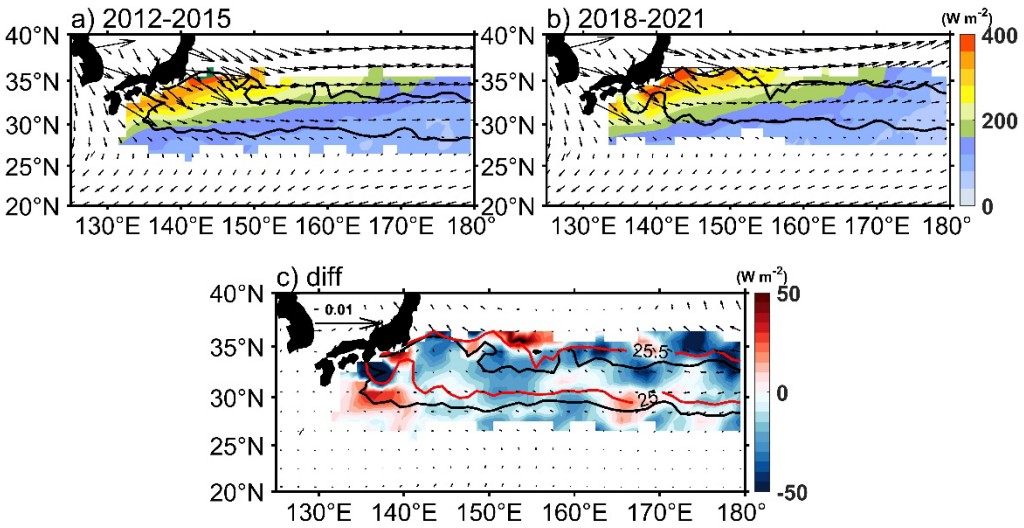

**Figure 14:** The wintertime (January−March) surface wind stress (vectors; N m⁻²), SLP (white contours; hPa) and wind stress curl (color shading; N m⁻³) in North Pacific Ocean from ERA5 in (a) 2012−2015 and (b) 2018−2021; (c) the difference between (b) and (a) (i.e., (b) minus (a)). The outcropping lines of $\sigma_\theta$=25.0 kg m⁻³ (the southernmost one) and 25.5 kg m⁻³ (the northernmost one) are superimposed in the black solid (2012−2015) and red solid contours (2018−2021). The green box in the figure (c) indicates the central North Pacific (140°W−170°W).

**Figure 15:** The averaged -$Q_{net}$ (W m⁻²) from ERA5 for years (a) 2012−2015 and (b) 2018−2021 in winter. The solid lines bound the $\sigma_\theta$=25.0−25.5 kg m⁻³ outcrop region time averaged from January to March. (c) The difference (the values of 2018−2021 minus that of 2012-2015) from ERA5 in the outcrop region between the wintertime -$Q_{net}$ (W m⁻²) in the 2018−2021 and that in 2012−2015 (positive values represent the larger ocean heat loss). The outcropping lines of $\sigma_\theta$=25.0 kg m⁻³ (the southernmost one) and 25.5 kg m⁻³ (the northernmost one) are superimposed in the black solid (2012−2015) and red solid contours (2018−2021). The wintertime surface wind stress (vectors; N m⁻²) and difference are also superimposed from ERA5.

First, the recently observed NPSTMW volume loss appears closely related to the intensity of the prevailing westerlies during the period of PDO phase shift. During 2018−2021, the westerlies occur between 30°N and 50°N, with the AL residing to the north over the subpolar gyre (Figure 14c). The intensity of the westerlies weakens in the western part of the basin (Figure 14). The northward shift of the westerlies causes a weaker wind to its south in the area of NPSTMW formation and subduction (Figure 15c). Consequently in 2018−2021, a persisting weakening of the intensity of wintertime surface wind stress is led over the outcropping window (Figure 15c). During the cold season, weakening of the overlying wind stress can subsequently prohibit the local heat loss in the NPSTMW formation region (Bond and Cronin, 2008). Meanwhile, the monthly time-depth section of temperature, stratification ($N^2 = -\dfrac{g}{\rho}\dfrac{\partial \sigma_\theta}{\partial z}$) anomalies relative to the 2004−2021 climatology and MLD averaged within the ventilation region is indicated in Figure 16. The intensified preconditioning of near surface stratification (<150 m)

in the warm season since 2018 (Figure 17) has increased due to the remarkable near sea surface warming (Figure 16). The positive temperature anomalies during 2018−2021 are related to the positive net sea surface flux anomalies and the vertical entrainment diagnosed in the MLT heat budget (Figures 12). This may further inhibit the necessary wintertime convection (i.e., the origin of the deep MLD in winter) essential for new NPSTMW formation in the following winter (Tomita et al., 2010).

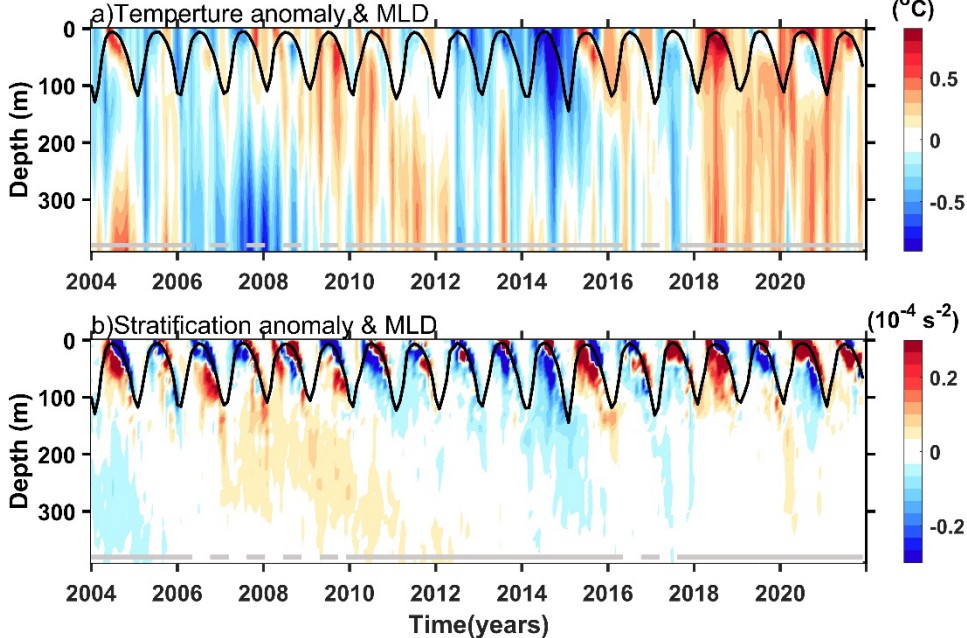

**Figure 16: Monthly time series of (a) temperature anomaly (ºC) relative to the 2004−2021 climatology, (b) stratification anomaly (10⁻⁴ s⁻²) relative to the 2004−2021 climatology and MLD (black lines) obtained from RGA dataset averaged over the NPSTMW formation region. Solid (dashed) bars indicate stable (unstable) periods of the KE.**

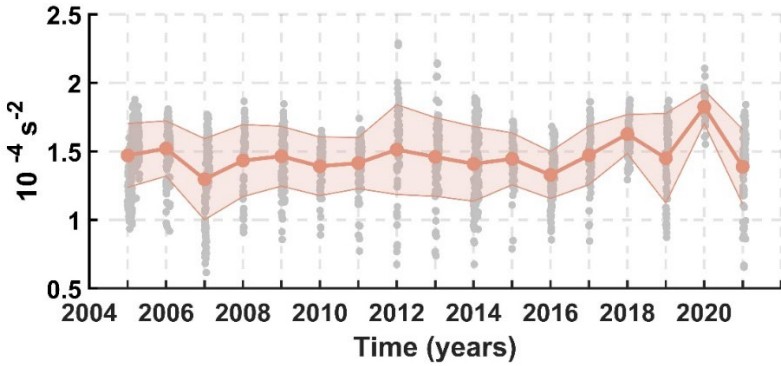

**Figure 17: The time series of near surface (<150 m) stratification during the warm season (July−September). Gray circles represent values observed from individual Argo profiles in the RG region. The circles-line represents mean values and the shading around each line shows the standard deviation.**

Indeed, as a result of the poleward shift of the westerlies, local easterly wind anomalies are shown over the NPSTMW formation region during the winter of 2018−2021 (Figure 15c), which cause the northward Ekman transport. The anomalous

northward Ekman advection of shallow ML with lighter densities would cause the northward migration of the NPSTMW outcropping lines (Figure 18c). Thus, the suppressed local heat loss and northward migration of lighter densities from south are coherent with the appearance of wintertime surface lighter density water mass (Figure 18).

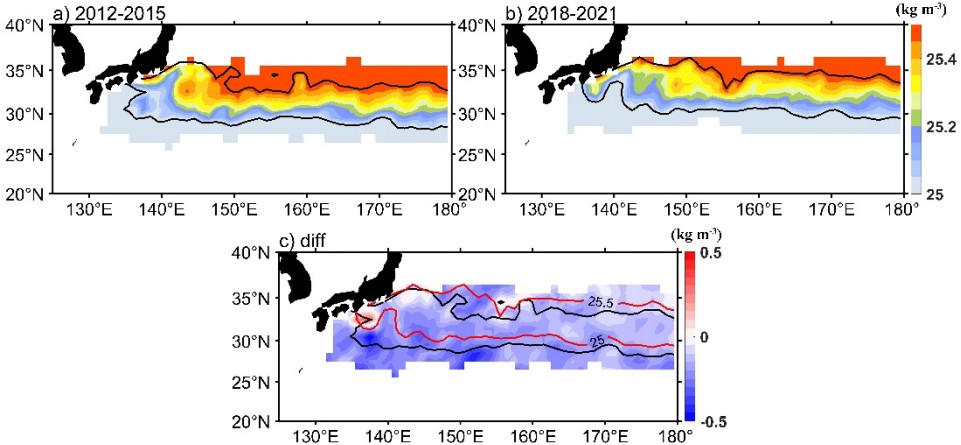

**Figure 18: The averaged surface potential density (kg m$^{-3}$) for years (a) 2012−2015 and (b) 2018−2021 in winter. The solid lines**
**bound the $\sigma_\theta$ =25.0−25.5 kg m$^{-3}$ outcrop region time averaged from January to March. (c) anomalies of surface potential density for years 2018−2021, as a deviation from the wintertime time mean (over 2012−2015) of surface potential density, in kg m$^{-3}$. The outcropping lines of $\sigma_\theta$ =25.0 kg m$^{-3}$ and 25.5 kg m$^{-3}$ are superimposed in the black solid (2012−2015) and red solid contours (2018−2021).**

## 4 Summary and Discussion

Though the KE jet is in a persistent stable state during a recent time of 2018−2021 which is somewhat analogous to the stable KE state in 2012−2015, the anomalous decrease of the NPSTMW volume and density in 2018−2021 have been observed from the Argo observation data. This is particularly pronounced in the subtropical RG region during winter and is most evident near the KE bounded by 30°N−34°N, 141°E−153°E. The mean NPSTMW volume drop by ~21% during 2018−2021 relative to 2012−2015 in the whole NPSTMW formation region. The density decrease is presented as the loss of

NPSTMW volume in denser density range (approximately $\sigma_\theta$ >25.2 kg m$^{-3}$) since 2018.

Although the KE is in a persisting stable state of 2018−2021, the NPSTMW volume loss is found in an analysis period when the PDO changes from its positive phase to a negative phase. The correlation between NPSTMW volume variability and the PDO index suggests that the NPSTMW variability is link to the modulation of the basin-scale atmospheric forcing. We can be enlightened how these mechanisms may play a role in interannual NPSTMW variability through an identification of

analogous cases in the precedent studies. Previous studies have linked the KE dynamic state changes to the basin-scale wind

pattern. For more than two decades since 1993, the decadal variability of NPSTMW has been modulated by the decadal variability of the KE system associated with the PDO and the westward propagation of the MTD anomalies in the central North Pacific (Qiu and Chen 2006; Rainville et al. 2014; Oka et al. 2015; Cerovečki and Giglio 2016). However, the distant effect is the first baroclinic Rossby wave as the primary message of wind variability from the central Pacific to the NPTMW formation region over a time lag of ~4 years (Sugimoto & Hanawa, 2010). The variabilities of MTD anomalies and KE state due to PDO (a ~4-yr lag) are corresponding to the propagation speed of the first baroclinic Rossby wave. However, Davis et al. (2010) also demonstrated that an interannual signal of the NPSTMW volume variability is also correlated with the PDO with zero time, which implies the connection of NPSTMW to the basin-scale ocean circulation. With this, modulations of upper-ocean properties driven by the varying strength and the position of the westerlies as well as the regional air-sea heat flux pattern are recognized as the significant contributions to the variability of NPSTMW volume on interannual time scales. The results presented in this study shows that there is no significant time lag of the distant effect between NPSTMW variability and the change of basin-scale wind stress pattern (i.e., the PDO phase shift). Meanwhile, the current stable KE state seems to have begun with the initiation of LM (Qiu et al., 2020), it has also been supported by basin-wide wind forcing (Qiu et al., 2023). This suggests that different mechanisms account for the recent NPSTMW decrease on interannual time scales, which resembles the theoretical framework in Davis et al. (2010) not in Qiu and Chen (2006) and Sugimoto & Hanawa (2010). Over the 2018−2021, the interannual signal is evident in the changes of NPSTMW volume, density and outcropping isopycnals moving, with this signal determined by the yearly difference in NPSTMW formation and subduction. The distinct interannual signal of NPSTMW in 2018-2021 is correlated well with PDO changes from its positive phase to a negative phase in recent years. It is supported by studies of the formation and subduction rate in the northwestern Pacific. In 2018−2021, the surface formation rate intensely declines in the upstream of KE (141°E-153°E). Meanwhile, in 2018−2021 only the lighter variety of water in the NPSTMW density range is replenished by surface formation. The total subduction volume of NPSTMW in the ventilation region subsequently has declined by 37% on average. Furthermore, the interannual variability of NPSTMW subduction is directly linked to the ML change. Such insufficient development of wintertime ML in the ventilation region of Northwestern Pacific and decline of NPSTMW formation are associated with the less heat loss and large-scale wind stress change in 2018−2021. This result is in agreement with the negative phase of the PDO shift studied here.

Although occurred locally, interannual variability of NPSTMW is intensely associated with the shift of basin-scale wind pattern. Based on this work, the dynamic impact of large-scale wind modification on the recent NPSTMW changes has been further examined. A scenario for these correlations of the physical causality is that (1) The northward shift of the westerlies when the PDO shift from a positive to a negative phase in 2018−2021 produces a weaker wintertime wind stress over the NPSTMW formation region, resulting in less heat loss to the atmosphere and shallower wintertime ML. Furthermore, the annual subduction dominated by lateral induction and temporal induction subsequently decline largely associated with wintertime ML change. (2) Diagnostic calculation in ML heat budget indicates that the change of the upper ocean warming is the key factor in determining the MLD variations in the NPSTMW formation region. The recent decrease of air-sea heat

exchange and the vertical entrainment act to weaken the vertical mixing and decrease the MLD, resulting in the weakening of subduction. (3) The air-sea heat exchange and the vertical entrainment are both associated with the overlying atmosphere. As a result of the poleward shift of the westerlies, easterly wind anomalies in the NPSTMW formation region cause the northward Ekman transport of the shallower ML with lighter densities from south. The significantly intensified preconditioning of near surface stratification (<150 m depth) associated with near surface warming is also a relevant factor.

They restrain wintertime convection and deepening of the ML, resulting in reduced transformation and formation of water in the NPSTMW density range.

The salient contribution of this study is that it reveals the anomalously decreasing NPSTMW in 2018−2021 is strongly associated with a response to a PDO phase shift event. We believe the that results of this study will be a step toward the deep understanding of the relationship between NPSTMW interannual variations and modulation of basin-scale wind pattern in

recent years. Owing to the NPSTMW wide range of influence, the present study may have broad implications for climate change and biogeochemical cycles (Holbrook et al., 2019; Kuroda and Setou, 2021; Tak et al., 2021).

**Data availability**

The gridded Argo product was obtained from Roemmich-Gilson Argo Climatology (https://sio-argo.ucsd.edu/RG_Climatology.html). The monthly mean net heat flux, sea level pressure and wind stress products are

535 obtained from ECMWF-ERA5 (https://doi.org/10.24381/cds.f17050d7). The sea surface height anomaly is provided by CMEMS (https://doi.org/10.48670/moi-00148, https://doi.org/10.48670/moi-00149).

The MOAA-GPV datasets is provided by the Japan Agency for Marine-Earth Science and Technology (https://www.jamstec.go.jp/argo_research/dataset/moaagpv/moaa_en.html).

**Author contributions**

JS and CL planned the study. JS, CL and YZG designed the analysis framework. JS processed data, conducted the analysis, and wrote the paper. ZN collected the data. CL, YZG, FGZ and PLL contributed with the analysis performance and interpretation of the results. JS and CL revised and edited the final version of the paper.

**Competing interests**

The contact author has declared that none of the authors has any competing interests.

## Disclaimer

Publisher's note: Copernicus Publications remains neutral with regard to jurisdictional claims in published maps and institutional affiliations.

## Acknowledgments

The Argo Program is part of the Global Ocean Observing System. These data were collected and made freely available by the International Argo Program and the national programs that contribute to it (http://www.argo.ucsd.edu; http://argo.jcommops.org). Dr. Eitarou Oka provided useful and constructive comments that improved the manuscript. The comments of two anonymous reviewers were helpful in revising the manuscript.

## Financial support

The work was jointly supported by the National Natural Science Foundation of China (grant no. 42206003), the Scientific and technological projects of Zhoushan (2022C81010), the Major program of Pilot National Laboratory for Marine Science and Technology (Qingdao) (2022QNLM040004-3), the Finance science and technology project of Hainan province (ZDKJ202019), and the National Science Foundation of China (grant no. 42176016).

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
