# Peer review of "North Pacific Subtropical Mode Water Volume and Density Anomalous Decrease in a Recent Stable Kuroshio Extension Period from Argo Observations"

_EGUsphere, 2023_

## Author Comment (AC1)

**Response to reviewer**

We would like to deeply thank the reviewer for constructive comments which greatly improve the manuscript. The followings are detailed response to each comment. Corresponding modifications are highlighted in the new submission.

To Reviewer 1,

*(1) As cited in the introduction, Oka et al. (2021) investigated NPSTMW using Argo floats observations. They showed that annual-mean volume actually increased in the northeast region (28–40ºN, 140ºE–170ºW) from 2017 to 2019, while that in the other region decreased. The northeast region includes the main formation region of NPSTMW with densities of larger than 1025.2 kg/m3. This is important difference and needs the comparison with Oka et al. (2021).*

**Response**: We thank the reviewer's valuable comment. According to the reviewer's suggestion, we have added the comparison with Oka et al. (2021) in the section 3.1 of the new submission (L216-220): Oka et al. (2021) investigated NPSTMW change in the region (15ºN–40ºN, 120ºE–170ºW) using Argo floats observations. They showed that annual-mean volume increased lightly in the northeast region (28ºN–40ºN, 140ºE–170ºW) from 2017 to 2019, while that in the other region decreased significantly. In fact, the NPSTMW in the whole region have gradually decreased in recent years. Hence, the recent decline of NPSTMW observed in RGA dataset is definitely consistent with the results of Oka et al. (2021).

*(2) As shown by Qiu et al. (2020) and Qiu (2023), the stable KE and its northward shift is a combined response to the Kuroshio LM south of Japan and the Rossby wave forcing from the interior wind stress across the North Pacific. The KE northward shift can affect the overlying storm track and basin-scale wind field (Qiu et al. 2020). Is the northward shift of the westerlies, shown by the authors, associated with the stable KE? Or is it due to the PDO? Some additional analyses will be necessary. In addition, are the surface warming and intensified stratification also due to the northward shift of the KE?*

*Qiu et al. (2020), JC, doi: 10.1175/JCLI-D-20-0237.1*

*Qiu (2023), Oceanography in Japan, doi: 10.5928/kaiyou.32.3-4_67.*

**Response**: We thank the reviewer's valuable comment. The current stable KE state seems to have begun with the initiation of LM (Qiu et al., 2020), it has also been supported by basin-wide wind forcing (Qiu et al., 2023). The dominant surface wind stress forcing in the midlatitude North Pacific basin is related to the PDO phases (Qiu et al., 2023) (L414-418). The average of temperature tendency anomaly during the cooling season of 2018-2021 (Figure 12b) is positive. It is contributed by the air-sea heat exchange (38.0%), the vertical entrainment through the base of the ML (37.0%), the Ekman advection (17.6%), and the geostrophic advection (7.4%). This result demonstrates that, in the NPSTMW formation region, the weak processes of the air-sea heat exchange and the vertical entrainment play an important role in the ML warming during 2018−2021. This result is demonstrated in section 3.4 of the new submission through the ML heat budget analysis (L359-388).

Qiu, B., Chen, S., Schneider, N., Oka, E., and Sugimoto, S.: On the reset of the wind-forced decadal Kuroshio Extension variability in late 2017, J. Clim., 33, 10813–10828, 2020.
Qiu, B., Chen, S., and Oka, E.: Why did the 2017 Kuroshio large meander event become the longest in the Past 70 years?, Geophys. Res. Lett., 50, e2023GL103548, 2023.

*(3) Davis et al. (2011) showed no correlation between NPSTMW and wintertime surface cooling in the RG region, consistent with Qiu and Chen (2006). However, the decline in the NPSTMW volume after 2018 seems to be caused by wintertime surface cooling. Please discuss the correspondence with Davis et al. (2011) and Qiu and Chen (2006).*

**Response**: Thanks. We have discussed the correspondence with Davis et al. (2011) and Qiu and Chen (2006) in the section 4 of the new submission (L486-500): For more than

two decades since 1993, the decadal variability of NPSTMW has been modulated by the decadal variability of the KE system associated with the PDO and the westward propagation of the MTD anomalies in the central North Pacific (Qiu and Chen 2006; Rainville et al. 2014; Oka et al. 2015; Cerovečki and Giglio 2016). However, the distant effect is the first baroclinic Rossby wave as the primary message of wind variability from the central Pacific to the NPTMW formation region over a time lag of ~4 years (Sugimoto & Hanawa, 2010). The variabilities of MTD anomalies and KE state due to PDO (a ~4-yr lag) are corresponding to the propagation speed of the first baroclinic Rossby wave. However, Davis et al. (2010) also demonstrated that an interannual signal of the NPSTMW volume variability is also correlated with the PDO with zero time, which implies the connection of NPSTMW to the basin-scale ocean circulation. With this, modulations of upper-ocean properties driven by the varying strength and the position of the westerlies as well as the regional air-sea heat flux pattern are recognized as the significant contributions to the variability of NPSTMW volume on interannual time scales. The results presented in this study shows that there is no significant time lag of the distant effect between NPSTMW variability and the change of basin-scale wind stress pattern (i.e., the PDO phase shift). Meanwhile, the current stable KE state seems to have begun with the initiation of LM (Qiu et al., 2020), it has also been supported by basin-wide wind forcing (Qiu et al., 2023). This suggests that different mechanisms account for the recent NPSTMW decrease on interannual time scales, which resembles the theoretical framework in Davis et al. (2010) not in Qiu and Chen (2006) and Sugimoto & Hanawa (2010).

*(4) L173-176. It is hard to see the changes from Fig. 4a. In addition, are these changes statistically significant?*

**Response**: Thanks. Following reviewer's suggestion, we also conducted the Mann-Kendall test on the NPSTMW volume in denser density range (approximately $\sigma_{\theta} > 25.2$ kg m$^{-3}$) from 2012 to 2018 following the method of Xu et al. (2022). For the present study, the null hypothesis of no trend is rejected when the absolute value of the standard

deviation $|Z|>1.96$ at the significance level is 0.05. Yes, these changes are statistically significant. There is a decreasing trend for the change of the NPSTMW volume in denser density range ($Z=-3.09$). Particularly, the mean NPSTMW volume in denser density range (approximately $\sigma_\theta$>25.2 kg m$^{-3}$) appears to decrease from $11.07\times10^{13}$ m$^3$ in 2012−2015 to 8.77 $\times10^{13}$ m$^3$ in 2018−2021. L220-225 in the new submission

Xu, L., Wang, K., and Wu, B.: Weakening and Poleward Shifting of the North Pacific Subtropical Fronts from 1980 to 2018, J. Phys. Oceanogr., 52, 399–417, https://doi.org/10.1175/JPO-D-21-0170.1, 2022.

*(5) L219-228. The differences between 2012-2015 and 2018-2021 are not clear from Fig. 7. Please revise the figures and show the significance together.*

**Response**: Thanks. We have revised Figure 7 and discussion of this Figure (L274-287).

*(6) Please check abbreviations and if they are used throughout the text or if they are not defined more than once. For example, the mixed layer is defined as ML at L33, but later it is still used (e.g., L138). PV is defined at L26 and 55. RG dataset (L89) and RG region (L113). The same abbreviation is a bit confusing.*

**Response**: Thanks. We have checked abbreviations and used RGA dataset and RG region in the new submission (e.g., L99, L126).

*(7) L136. What do the authors cite here by Oka et al. (2011)?*

**Response**: Sorry for the incorrect reference in original manuscript. Modified (L616-617). We computed the horizontal geostrophic velocity field **u** relative to a reference level of 1000 m following the method of Oka et al. (2011).

Oka, E., Suga, T., Sukigara, C., Toyama, K., Shimada, K., and Yoshida, J.: "Eddy Resolving" Observation of the North Pacific Subtropical Mode Water, J. Phys.

Oceanogr., 41, 666–681, https://doi.org/10.1175/2011JPO4501.1, 2011.

*(8) L136. Does ERA5 provide wind stress data? Do the authors calculate wind stress from the surface wind data?*

**Response**: Yes, ERA5 provides wind stress data. No, we used the surface 10 m wind stress data provided by ERA5.

*(9) Equation 6. Is M a function of not only sigma theta but also time?*

**Response**: Yes, we have corrected the expression of $M(\sigma_\theta, t)$ instead of $M$ (L176).

*(10) Section 3.1. Please show a stable and unstable KE period together with Fig 4.*

**Response**: Thanks, we have added the stable and unstable KE period into the Fig 4 in the new submission (L244).

*(11) L169-170. What do authors mean by "Except for … "?*

**Response**: We are sorry for the poor expression of this sentence in the original manuscript. We mean, "In a short time period of 2006-2009 when the KE jet is unstable, the NPSTMW volume has a dramatic decrease" (L211-212).

*(12) Figures 5 and 7. Winter average, not annual average?*

**Response**: Winter average (L273, L286).

*(13) L200-201. Do Suga and Hanawa (1990) pointed out these things?*

**Response**: No, we are sorry for the incorrect reference. This reference has been removed here.

*(14) L207-208. This sentence seems to be different from what the authors say at L199-200.*

**Response**: Thanks. We have removed this incorrect sentence.

*(15) L242-244. The northward shift is obvious in not only the dense isopycnal but also the light isopycnal.*

**Response**: Thanks, we have corrected the expression of this sentence as: The surface warming due to the northward KE shift may cause the northward movement of the outcropping isopycnals in the NPSTMW density range and inhibit the deepening of the ML south of KE in the winter of 2018-2021 (L301-303).

*(16) L252. What is RGA?*

**Response**: Sorry for the incorrect abbreviation. RGA is the abbreviation of Roemmich and Gilson Argo dataset in the new submission (L99).

*(17) L262. What is the ventilation region here? Is this the RG region?*

**Response**: The ventilation region here is the whole analysis region 125°E-180°, 20°N-40°N, not the RG region (L332-334).

*(18) L264-265. I am not sure if this is supported by Fig. 9.*

**Response**: Thanks. This sentence is moved.

*(19) L273. "Except for" should be "In addition to"*

**Response**: Thanks. Changed (L390).

*(20) Fig. 12. Why is the wind stress curl masked along the international dateline?*

**Response**: Sorry for this poor expression of Fig. 12. Fig. 12 is modified in the new submission (L428-433).

*(21) Fig. 12. What do the authors mean by ensembled? Do the authors use ERA5 ensemble mean data?*

**Response**: we do not use ERA5 ensemble mean data. Sorry for this incorrect expression.

"ensembled" is removed (L429-433).

*(22) L308. It is hard to see the change in the westerlies from Fig. 6. Instead of the wind stress curl, the authors may want to show the magnitude of the wind stress.*

**Response**: Thanks. We have added the wind stress vector in Figure 15 of the new submission (L434-440).

*(23) L314-315. The intensification of the surface stratification seems to be different from what we expect from the deepening of the main thermocline in Figure 11. Is the stratification indeed weakened in the depth of the thermocline after 2018?*

**Response**: Yes, it also proved by the raw Argo in the RG region. The stratification indeed weakened in the depth of the thermocline after 2018 as shown in the Figure 20 of the new submission (L459-462).

*(24) In addition, I would like to suggest the authors to ask native speakers of English to check and correct the English language usage. It is sometimes difficult to follow correctly what is mentioned.*

**Response**: Thanks. We have check and correct the English language usage following suggestion.

---

## Author Comment (AC2)

**Response to reviewer**

We would like to deeply thank the reviewer for constructive comments which greatly improve the manuscript. The followings are detailed response to each comment. Corresponding modifications are highlighted in the new submission.

**To Reviewer 2,**

1. It should be emphasized a little more that the switch to the KE stable state in 2017 is not wind-forced but is a response to an occurrence of the Kuroshio large meander, whereas the stable KE path during 2012–2015 results from wind stress forcing (Qiu et al., 2020). In other words, although the KE is in the stable state in both periods of 2012–2015 and 2018–2021, backgrounds are not necessarily the same.

**Reference:**

1. Qiu et al. (2020): On the reset of the wind-forced decadal Kuroshio Extension variability in late 2017. Journal of Climate, 3, 10813-10828. doi:10.1175/JCLI-D-20-0237.1

**Response**: Thanks. We would expect more detailed explanation here for readers' understanding (L77-82). We have modified the expression in introduction as "In August 2017, KE switched from an unstable state to a stable state in association with the occurrence of the Kuroshio large-meander (LM) path south of Japan (Figure 1), although negative SSH and MTD anomalies associated with the positive PDO phase were arriving from the central North Pacific (Qiu et al. 2020). Since then, Kuroshio LM and the stable state of the KE have lasted for more than six years (Qiu and Chen, 2021; Qiu et al., 2023; Usui, 2019), while the NPSTMW volume has declined (Oka et al., 2021). It is worth noting that the current stable KE state seems to have begun with the initiation of LM (Qiu et al., 2020), it has also been supported by basin-wide wind forcing (Qiu et al., 2023)."

2. An impact of wind-induced stratification changes on the STMW formation should be carefully considered. Figure 11a shows Pacific decadal oscillation (PDO) is positive during 2014–2017. This means that upper ocean stratification in the central North Pacific is intensified (i.e., shoaling of the main thermocline is caused). Such signals propagate westward as oceanic Rossby waves and affect the STMW formation region during 2018–2021, considering a 4-year lag suggested by a previous study (Sugimoto and Kako, 2016 cited in the present study). According to Sugimoto and Kako (2016), this may also influence the STMW formation decline during 2018–2021 by hindering deep convection and then development of winter Mixed layer.

Response: Thanks. The impact of wind-induced subsurface stratification changes (i.e., the changes of MTD) on the STMW formation is carefully considered in the section 3.5 of the new submission following the reviewer's the suggestion (L393-404): The NPSTMW volume variations in 2006-2009 (2012-2015) (Figure 4a) are closely controlled by the MTD because the PDO-related stronger (weaker) subsurface stratification from the seasonal thermocline to the main thermocline propagated from the central North Pacific (Figure 13). Meanwhile, Figure 16 also shows the distant effects of MTD (i.e., subsurface stratification) change on NPSTMW formation in the western North Pacific with some time lag (about 3-4 years). Although PDO is positive during 2014–2017, such signals of negative MTD, which propagates westward as oceanic Rossby waves, vanishingly affect the NPSTMW formation region due to the persisting strong sea surface warming (with positive SSH anomaly as its proxy) in NPSTMW formation region during 2018–2021 (Figure 13). However, since 2018, the PDO has transitioned from a positive to negative phase (Figure 13a), with the concomitant positive SSH anomaly and deepened MTD (Figures 13b and 13c). The positive MTD anomaly which weakens the background stratification should have facilitated the development of wintertime MLD and NPSTMW formation, as suggested as that in 2012-2015. On the contrary, the MLD and the subduction of NPSTMW do not have such sufficient developments in the late winter of 2018-2021 (Figures 8 and 9).

3. The authors focused mainly on formation/volume (and density) of STMW. I suggest that STMW temperature should also be intensively examined as well. According to previous studies (and Figure 13a in the present study), temperature of subducted STMW (i.e., summertime STMW) tends to increase (decrease) during the unstable (stable) KE when the winter mixed layer becomes relatively shallow (deep) (e.g., Oka et al., 2019 cited in the text; see their Figure S5). However, Figure 13a of the present study indicated different feature of STMW temperature between 2012–2015 and 2018–2021. This is also remarkable and is related to the interest of this study. Comprehensive analyses not limited to STMW formation/volume would give us much deeper understandings of STMW.

**Response**: We thank the reviewer's valuable comment. We examined the core layer potential temperature (CLT) of the NPSTMW using the ML heat budget analysis in section 3.4 of the new submission as well following reviewer's suggestion (L341-388).

4. Descriptions are too focused on periods of 2012–2015 and 2018–2021. Although I understand that the authors have an interest in STMW in 2018–2021 and compare it with that in 2012–2015, discussions can be started from overall features. Explanation with a wider view would be better.

**Response**: Thanks. Discussions with a wider view has also been added into sections of results following reviewer's suggestion below in the new submission (e.g., L229-236, L359-375).

5. I concern that STMW in 2018–2021 is compared with only one period of the KE stable state (2012–2015). Besides 2012–2015, the KE is in the stable state during 2003–2005. If the authors use another gridded dataset using Argo float data (MOAA-GPV; Hosoda et al., 2008) in addition to data by Roemmich and Gilson (2009), it is possible to further compare STMW formation/volume in 2018–2021 with another KE stable period. Although the number of Argo float profiles is not large in early-2000s, a

previous study discussed STMW using Argo float profiles since 2003 (Sugimoto et al., 2013).

Reference:

1. Hosoda et al. (2008): A monthly mean dataset of global oceanic temperature and salinity derived from Argo Float observations. JAMSTEC Report Research and Development, 8, 47-59. doi:10.5918/jamstecr.8.47

2. Sugimoto et al. (2013): Marked freshening of North Pacific subtropical mode water in 2009 and 2010: influence of freshwater supply in the 2009 warm season. Geophysical Research Letters, 40, 3102-3105. doi:10.1002/grl.50600.

**Response**: Thanks. Following reviewer' suggestion, we have compared NPSTMW volume in 2018–2021 with that in 2003–2005 using MOAA-GPV dataset in section 3.1 of the new submission (L229-236, L253-258).

6. The authors pointed out importance of Ekman transport in the text. But Ekman heat advection is not discussed. It can be calculated from wind stress and temperature data, and further its effect may be considered in eq (5).

**Response**: Thanks. We have discussed Ekman heat advection in the new submission (L424-425): The horizontal heat advection rate by the Ekman velocity  $(-(\mathbf{u}_e \cdot \nabla T_m))$  has correspondingly declined by 16% during 2018-2021 relative to 2012-2015. Through the ML heat budget analysis in section 3.4 (L359-369), we find: The temperature tendency in climatological average is counterbalanced by the air-sea heat exchange (41.2%), the vertical entrainment through the base of the ML (26.7%), the Ekman advection (8.2%), and the geostrophic advection (23.9%). Thus, the temporal variability of MLT in the cooling season is dominated by the anomalies of the air-sea heat exchange, the vertical entrainment and the geostrophic advection in climatological average. The contribution of the Ekman advection to the annual temporal variability of

MLT in the cooling season is so small. Finally, the effect of Ekman heat advection is not needed to considered in Eq (5).

7. Timeseries of winter NHF and heat flux due to Ekman transport in the STMW formation region is worth discussing and being compared with late-winter MLD and subduction volume.

**Response**: We thank the reviewer's valuable comment. We have added the ML heat budget analysis in section 3.4 and section 3.5 of the new submission (L370-388, L418-427, L434-440). Timeseries of winter  $Q_{net}$  and heat flux due to Ekman transport in the STMW formation region have been discussed and being compared with late-winter MLD and subduction volume.

8. How about to add horizontal bars or something to panels of timeseries in Figures 4, 9, 10, 11, and 13 in order to show periods of stable/unstable states of the KE? Further, in those panels, average values of 2012–2015 and 2018–2021 is better to be plotted to be compared easily because their difference is focused throughout the present study. **Response**: Thanks. Modified following reviewer's suggestion (Figures 4, 9, 10, 13, and 16 in the new submission).

**9. L12 and many places**

"-" instead of "-".

**Response**: Thanks. Modified (e.g. L12 and so on).

10. L31.

To explain a role of STMW as the memory of atmospheric conditions, it would be better to mention the reemergence of sea surface temperature anomalies (e.g., Hanawa and Sugimoto, 2004; Sugimoto and Hanawa, 2005).

Reference:

1. Hanawa and Sugimoto (2004): 'Reemergence' areas of winter sea surface temperature anomalies in the world's oceans. Geophysical Research Letters, 31, L10303. doi:10.1029/2004GL01990

2. Sugimoto and Hanawa (2005): Remote reemergence areas of winter sea surface temperature anomalies in the North Pacific, Geophysical Research Letters, 32, L01606. doi:10.1029/2004GL021410.

Response: Thanks. Modified following reviewer's suggestion (L34-38).

11. L59

"Cerovečki and Giglio (2016)" Instead of "(Cerovečki and Giglio, 2016)". Response: Thanks. Modified (L65-66).

*12. L70-73*

Information written in the text is not up-to-date. The recent Kuroshio large meander persists for more than six years and is the longest record in the past 70 years (Qiu et al., 2023). Even if the present study would examine STMW before 2022, this should be written correctly.

Reference:

1. Qiu et al. (2023): Why did the 2017 Kuroshio large meander event become the longest in the past 70 years? Geophysical Research Letters, 50, 32023GL103548.

*doi:10.1029/2023GL103548*.

Response: Thanks. Modified (L79-80).

13. L83 (Figure 1 caption)

"Copernicus Marine Environment Monitoring Service (CMEMS)" instead of "CMEMS". Figure 1 is referred to at L70, but full name description of the CMEMS in the text is at L99.

Response: Thanks. Modified (L93).

14. L95, L158, L213 (Figure 6 caption), ...

Unify the abbreviation of net surface heat flux (HF or NHF). **Response**: Thanks. Modified as  $Q_{net}$  (L108, L171, L435).

15. L103

"RG dataset" and "RG region" are so confusing. I suggest to use other words. Further, it is not appropriate that the abbreviation of RG region is used in the caption of Figure 2 (referred at L94) before it is defined in the main text (at L113).

**Response**: Thanks. We have checked abbreviations and used the abbreviation of RGA dataset and RG region in the new submission (L99). We have modified the abbreviation of RG region is used in the caption of Figure 2 (L117).

16. L141

Declare what is Q in the text explicitly.

Response: Thanks. We have declared as "the PV of subducted water is related to the

subduction rate" in the text explicitly (L152).

17. L169-170

**I cannot understand this sentence.**

**Response**: We are sorry for the poor expression of this sentence in the original manuscript. We mean, "In a short time period of 2006-2009 when the KE jet is unstable, the NPSTMW volume has a dramatic decrease" (L211-212).

18. L199

Does the "the averaged area of the NPSTMW outcrop window" mean the "the averaged formation area of the NPSTMW"? If so, write so somewhere. **Response**: Yes. Modified (L263-264).

19. L199

I felt that descriptions at L199-200 and at L207-208 contradict a little. The authors may want to improve the explanation.

**Response**: Thanks. We have removed the expression of sentence in L207-208 of the original manuscript.

20. L213, L240, ...

The word of "winter" and "wintertime" should be defined uniquely in the text at the first reference. In the current manuscript, two definitions (January–March and February–March) exist.

**Response**: Thanks. The word of "winter" and "wintertime" is defined as January–March in the text at the first reference (L120).

**21. L221-L223**

Show the value of transformation rates and/or their difference in the lighter NPSTMW range to support the result.

Response: Thanks. Modified (L281-283).

22. L230 (Figure 7 caption)

"(a), (c) Annually averaged surface transformation rate and ..." instead of "Annually averaged (a) and (b) surface transformation rate and". **Response**: Thanks. Modified (L286-287).

23. L223-

Does it mean that the STMW formation is concentrated on its lighter density range? If so, write so somewhere explicitly. Further, explain why the STMW formation is concentrated on the lighter range.

**Response**: Thanks. The Figure 7 and discussion of this figure have been modified (L274-287).

*24. L227-228*

Need more explanation.

**Response**: Thanks. The Figure 7 and discussion of this figure have been modified (L274-287).

25. L242-245

The KE has shifted northward during 1993–2021 (e.g., Wu et al., 2021 cited in the manuscript; Kawakami et al., 2023). I suggest to discuss a relationship to the KE shift.

Reference:

 Kawakami et al. (2023): Northward shift of the Kuroshio Extension during 1993–2021. Scientific Reports, 13, 16223. doi:10.1038/s41598-023-43009-w.
Response: Thanks. Modified (L300-303).

26. L254

Suggest to discuss the mean MLD distributions during winter of 2012–2015 and 2018–2021 rather than showing only their difference. **Response**: Thanks. Modified (L312).

27. L264

Subduction volume depends on MLD probably not only 2018–2021 but also throughout the analysis period. Although I recognize that the main target of the present study is STMW formation in 2018–2021, how about start a discussion from an overall relationship between MLD and STMW formation by performing a correlation analysis or something else?

**Response**: Thanks. Modified (L320-330).

28. L276

*"Figure 4b" instead of "Figure 2b".* **Response**: Thanks. Modified (L393). 29. L283

"Figures" instead of "Figure".

Response: Thanks. Modified (L404).

30. L285 (Figure 11)

It would be better to locate the panel (a) at the left-hand (or right-hand) side of panels (b) and (c) with rotation of 90° as done in Qiu et al. (2023) (mentioned above) so that readers can compare the PDO index and sea surface height (SSH) and main thermocline depth (MTD) anomalies easily.

Response: Thanks. Modified (L407).

*31. L285*

In panel (c), was the global-mean sea level rise due to freshwater flux and thermal expansion removed?

**Response**: Not removed in the original manuscript, but we have removed the steric height changes in the new submission (L197-207, L407).

32. L294

Specify where the central North Pacific is in Figure 12a. **Response**: Thanks. Specified in Figure 14c (L428).

33. L301 (Figure 12)

The Order of panels are unnatural. Consider to rearrange as (a) 2012–2015, (b) 2018–2021, and (c) their differences. Further, sea level pressure (SLP) can be

superimposed by contours. SLP helps the readers to find the Aleutian Low discussed in the text.

Response: Thanks. Modified (L428).

34. L313

Specify what is stratification anomaly. How was it calculated?

**Response**: Thanks. The values of stratification anomaly are relative to the 2004–2021 climatology (L448). It is calculated from  $N^2 = -\frac{g}{\rho} \frac{\partial \sigma_{\theta}}{\partial z}$ .

---

## Author Comment (AC3)

**Response to reviewer**

We would like to deeply thank the reviewer for constructive comments which greatly improve the manuscript. The followings are detailed response to each comment. Corresponding modifications are highlighted in the new submission.

To Reviewer 3 (Dr. Eitarou Oka),

*1. I understood the authors' story that less NPSTMW was formed in 2018-2021 compared to 2012-2015 due to less atmospheric cooling and the resultant smaller MLD. However, Figure 13a shows that negative temperature anomalies during 2012-2013 when the PDO index was negative as in 2018-2021. Isn't there a possibility that positive temperature anomalies during 2018-2021 are due to horizontal heat advection from the meandering Kuroshio south of Japan? Moreover, isn't this heat advection the major cause of the less formation of NPSTMW after 2018? This is the only major comment from me. Otherwise, the manuscript is relatively well written, and I recommend its publication in OS after the above issue is addressed.*

**Response**: We thank the reviewer's valuable comment and recommendation. Through the ML heat budget analysis, the average of temperature tendency anomaly during the cooling season of 2018-2021 (Figure 12b) is positive. It is contributed by the air-sea heat exchange (38.0%), the vertical entrainment through the base of the ML (37.0%), the Ekman advection (17.6%), and the geostrophic advection (7.4%). This result demonstrates that, in the NPSTMW formation region, the weak processes of the air-sea heat exchange and the vertical entrainment play an important role in the ML warming during 2018−2021. The weak temperature advection by the Ekman flow also makes contributions to the warming of local MLT. This result is demonstrated in section 3.4 of the new submission (L341-388).

In addition, although the KE is in a stable state in 2018-2021, the average of geostrophic advection anomaly is negative during 2018-2021, which does not tend to warm the wintertime MLT as in 2012-2015 (Figure 12b). Thus, we study the parts attributable to geostrophic advection anomaly (Eq.(R1), Figure R1). The geostrophic advection

anomaly is contributed by $-\left(\overline{\mathbf{u}}_g \cdot \nabla T_m'\right)$ (36.4%), $-\left(\mathbf{u}_g' \cdot \nabla \overline{T}_m\right)$ (32.8%), and $-\left(\mathbf{u}_g' \cdot \nabla T_m'\right)$ (30.8%). Even though the advection of mean temperature by the anomalous geostrophic flow ($-\left(\mathbf{u}_g' \cdot \nabla \overline{T}_m\right)$) is positive (negative) in the stable (unstable) KE state during 2004-2021, which is also pointed out in Qiu (2000), the averaged advection of anomalous temperature by the mean geostrophic flow ($-\left(\overline{\mathbf{u}}_g \cdot \nabla T_m'\right)$) and $-\left(\mathbf{u}_g' \cdot \nabla T_m'\right)$ has largely negative effect on geostrophic advection anomaly during 2018-2021 (Figure R1). It indicates that the decreasing temperature gradient ($\nabla T_m'$) in the winter causes the recent cooling of the sea surface temperature in the ventilation region (Figure R1). Thus, we think that the positive temperature anomalies during 2018-2021 are not due to horizontal heat advection from the meandering Kuroshio south of Japan.

[Figure]

**Figure R1: Yearly time series of the anomaly of values (positive value indicates that the term increases MLT) contributes to the geostrophic advection anomaly (sum term in this Figure) relative to the 2004－2021 climatology from October of the previous year to March (cooling season) in Eq. (R1). Values are averaged in the wintertime ventilation region of 141°E－180°, 30°N－34°N. Solid (dashed) bars indicate stable (unstable) periods of the KE.**

According the method of Toniazzo et al. (2010), the geostrophic advection anomaly can be decomposed into the attributable parts:

$$-\left(\mathbf{u}_g \cdot \nabla T_m\right)' = -\overline{\mathbf{u}}_g \cdot \nabla T_m' - \mathbf{u}_g' \cdot \nabla \overline{T}_m - \mathbf{u}_g' \cdot \nabla T_m' \tag{R1}$$

Here, Overbar denotes a climatological average. Prime represents the anomalous values relative to the climatological average.

Toniazzo, T., Mechoso, C. R., Shaffrey, L. C., and Slingo, J. M.: Upper-ocean heat budget and ocean eddy transport in the south-east Pacific in a high-resolution coupled model, Clim. Dyn., 35, 1309–1329, 2010.

Qiu, B.: Interannual variability of the Kuroshio Extension system and its impact on the wintertime SST field, J. Phys. Oceanogr., 30, 1486–1502, 2000.

*2. L33: south of the Kuroshio Extension (KE) -> south of the Kuroshio and the Kuroshio Extension (KE).*

**Response**: Thanks. Modified (L40).

*3. L70-72, "In recent years, KE is in a stable state associated with the Kuroshio large-meander (LM) path south of Japan (Figure 1). Although a persisting Kuroshio LM and the resultant stable state of the KE has already exceeded four years (Qiu and Chen, 2021; Usui, 2019), the NPSTMW volume has declined since 2018 (Oka et al., 2021)." I would expect more detailed explanation here for readers' understanding. I would write, "In August 2017, KE switched from an unstable state to a stable state in association with the occurrence of the Kuroshio large-meander (LM) path south of Japan (Figure 1), although negative SSH and MTD anomalies associated with the positive PDO phase were arriving from the central North Pacific (Qiu et al. 2020, JC). Since then, Kuroshio LM and the stable state of the KE have lasted for more than six years (Qiu and Chen, 2021; Qiu et al., 2023; Usui, 2019), while the NPSTMW volume has declined (Oka et al., 2021)." Note that the current stable KE state seems to have begun with the initiation of LM (Qiu et al., 2020), it has also been supported by basin-wide wind forcing (Qiu et al., 2023, GRL).*

**Response**: Thanks. Modified following reviewer's suggestion (L77-82).

*4. L95, "net surface heat flux (HF)": This notation is somewhat misleading in eq. (5) (L158) because it looks like a product of H and F. Consider using a single character.*

**Response**: Thanks. Modified as $Q_{net}$ (L118,L171).

*5. L108, "sigma-theta = 25.0-25.5 kg m-3": To what temperature range does this density range correspond? I am just curious if the warmest (lightest) variety of NPSTMW formed south of Japan, especially in the Kuroshio LM period (Nishikawa et al., 2023, JO), is included in the authors' analysis.*

**Response**: This density range in our study corresponds to the temperature range as 16-18 ℃, which is mainly formed in the wintertime ventilation region of 141°E−180°, 30°N−34°N. These are the relatively cold NPSTMW. Thus, the warmest (lightest) variety of NPSTMW (exceeding 19 °C) formed south of Japan, especially in the Kuroshio LM period (Nishikawa et al., 2023, JO), is not included in our analysis.

*6. L169-170, "Except for a short time of 2006-2009 when the KE jet is unstable, the NPSTMW volume has a dramatic decrease during 2006-2009.": I do not understand. Do the authors mean, "In a short time period of 2006-2009 when the KE jet is unstable, the NPSTMW volume has a dramatic decrease."?*

**Response**: Yes, thanks. Modified (L211-212).

*7. L221-223, "transformation rates … were greatly reduced.": Not obvious for me from Fig. 7a,c.*

**Response**: Thanks. The Figure 7 and discussion of this figure have been modified (L274-287).

*8. L223-225, "the annually averaged surface formation rates … density range": Not obvious for me from Fig. 7b,d.*

**Response**: Thanks. The Figure 7 and discussion of this figure have been modified (L274-287).

*9. L273, "Except for the KE dynamic state change,": I do not understand.  Maybe, "In addition to the KE dynamic state change,", although the KE dynamic state and "oceanic precondition" are not independent from each other.*

**Response**: Yes, thanks. Modified (L390-391).

*10. L311, "is leaded": "is led"?*

**Response**: Thanks. Modified (L445).

---

## Author Response (AR2)

**Response to reviewers**

We would like to deeply thank the reviewers for their constructive comments which greatly improve the manuscript. The followings are detailed response to each comment. Corresponding modifications are highlighted in the new submission.

To Reviewer 1,

*(1) The authors have addressed all my questions and made improvements. I appreciate the effort of the authors and recommend the paper to be accepted.*

*One thing you may want to consider is the heat budget analysis of the mixed layer in Figure 12 and Section 3.4, which is added in the revision. The residual term is generally larger than the tendency term in the period from 2018 to 2021, which makes all the description sound unconvincing. I think that the authors can retain most of the results without Figure 12. So, I recommend deleting Figure 12 and the heat budget analysis or moving them into the discussion section. In the latter case, the authors may need to carefully describe the results.*

**Response**: We thank the reviewer's valuable comment and recommendation. According to the reviewer's suggestion, we have moved the Figure 12 and the heat budget analysis in the original manuscript into the discussion section while carefully describing the results in the new submission (L460-473).

To Reviewer 2,

*1. The authors successfully addressed my comments. I appreciate author's efforts for the revision. However, I have still one concern as written below. I am so sorry for a new comment, but this point would be crucial for this study. The authors may want to address it.*

*Sugimoto and Hanawa (2014) clearly showed from Argo Float observations that warmer and lighter NPSTMW (with potential temperature of 19.0−19.5℃ and potential density of 24.5−25.0 kg m−3) than usual is formed when the Kuroshio south of Japan takes meander paths including the large meander path. So, in order to investigate the NPSTMW during the Kuroshio large meander, the authors must consider much lower density range (e.g. 24.5−25.6) than that focused in the present manuscript.*

*Reference:*

*Sugimoto and Hanawa (2014): Influence of Kuroshio Path Variation South of Japan on Formation of Subtropical Mode Water. Journal of Physical Oceanography, doi:10.1175/JPO-D-13-0114.1*

**Response**: We thank the reviewer's valuable comment and recommendation. Following the reviewer's suggestion, we have obtained the monthly time series of volume of NPSTMW in $\sigma_\theta$=24.5−25.5 kg m$^{-3}$, $\sigma_\theta$=25.0−25.5 kg m$^{-3}$ and $\sigma_\theta$ =24.5−25.0 kg m$^{-3}$ obtained from the RGA dataset (as shown in Figure R1). From Figures R1(a) and R1(b), the variations of NPSTMW in $\sigma_\theta$=25.0−25.5 kg m$^{-3}$ contributes mostly to the changes of NPSTMW in $\sigma_\theta$=24.5−25.5 kg m$^{-3}$ including the NPSTMW in much lower density range, which accounts for 86.5%. Meanwhile, the variation tendency of NPSTMW in $\sigma_\theta$=25.0−25.5 kg m$^{-3}$ is consistent with that of NPSTMW in $\sigma_\theta$=24.5−25.5 kg m$^{-3}$. However, the NPSTMW in $\sigma_\theta$=24.5−25.0

kg m$^{-3}$ (much lower density range) has not shown a significant tendency of decrease in 2018-2021 (Figure R1c) as the depiction in Figure R1(a) and R1(b).

In addition, the NPSTMW in $\sigma_\theta$=24.5−25.0 kg m$^{-3}$ (with potential temperature of 19.0−19.5℃) is formed mainly south of the Kuroshio to the west of 140°E, while the distribution of the NPSTMW in $\sigma_\theta$=25.0−25.5 kg m$^{-3}$ is concentrated in the east of 140°E (Sugimoto and Hanawa, 2014; Oka et al., 2021; Liu et al., 2017). Meanwhile, the NPSTMW in $\sigma_\theta$=24.5−25.0 kg m$^{-3}$, which is mostly formed in south of the Kuroshio to the west of 140°E, has no significant decrease in 2018-2021 (Figure R1c). The decline of the NPSTMW in $\sigma_\theta$=25.0−25.5 kg m$^{-3}$ in 2018-2021, which is consistent with the interannual variability of the NPSTMW in $\sigma_\theta$=24.5−25.5 kg m$^{-3}$ in 2018-2021, is most noteworthy to studies (Figures R1a and R1a). Ultimately, our studies pay much attention on the changes of the NPSTMW in $\sigma_\theta$=25.0−25.5 kg m$^{-3}$ which is mostly formed in the east of 140°E.

Thus, even though considering the much lower density range in $\sigma_\theta$=24.5−25.0 kg m$^{-3}$, the most of the results in current manuscript will remain unchanged. Finally, much lower density range ($\sigma_\theta$=24.5−25.0 kg m$^{-3}$) is not focused in the present manuscript.

[Figure]

Figure R1: Monthly time series of volume of NPSTMW in (a) $\sigma_\theta$=24.5−25.5 kg m$^{-3}$ (b) $\sigma_\theta$=25.0−25.5 kg m$^{-3}$ (c) $\sigma_\theta$=24.5−25.0 kg m$^{-3}$ obtained from the RGA dataset. Solid (dashed) bars indicate stable (unstable) periods of the KE.

Sugimoto, Shusaku, & Hanawa, K. (2014). Influence of Kuroshio Path Variation South of Japan on Formation of Subtropical Mode Water. Journal of Physical Oceanography, 44(4), 1065–1077. https://doi.org/10.1175/JPO-D-13-0114.1

Oka, E., Nishikawa, H., Sugimoto, S., Qiu, B., & Schneider, N. (2021). Subtropical Mode Water in a recent persisting Kuroshio large-meander period: part I—formation and advection over the entire distribution region. Journal of Oceanography, 77, 781–795.

Liu, Cong, Xie, S., Li, P., Xu, L., & Gao, W. (2017). Climatology and decadal variations in multicore structure of the N orth P acific subtropical mode water. Journal of Geophysical Research: Oceans, 122(9), 7506–7520.

https://doi.org/10.1002/2017JC013071